# LLM-Guided Diagnostic Evidence Alignment for Medical Vision–Language Pretraining under Limited Pairing

**Huimin Yan** [1]  **Liang Bai** [1]  **Xian Yang** [2]  **Long Chen** [3]

## Abstract

Most existing CLIP-style medical vision–language pretraining methods rely on global or local alignment with substantial paired data. However, global alignment is easily dominated by non-diagnostic information, while local alignment fails to integrate key diagnostic evidence. As a result, learning reliable diagnostic representations becomes difficult, which limits their applicability in medical scenarios with limited paired data. To address this issue, we propose an LLM-Guided Diagnostic Evidence Alignment method (LGDEA), which shifts the pretraining objective toward evidence-level alignment that is more consistent with the medical diagnostic process. Specifically, we leverage LLMs to extract key diagnostic evidence from radiology reports and construct a shared diagnostic evidence space, enabling evidence-aware cross-modal alignment and allowing LGDEA to effectively exploit abundant unpaired medical images and reports, thereby substantially alleviating the reliance on paired data. Extensive experimental results demonstrate that our method achieves consistent and significant improvements on phrase grounding, image–text retrieval, and zero-shot classification, and even rivals pretraining methods that rely on substantial paired data.

## 1. Introduction

Medical vision–language pretraining (VLP) aims to learn transferable multimodal representations by modeling the

[1]Institute of Intelligent Information Processing, Shanxi University, Taiyuan, China [2]Alliance Manchester Business School, The University of Manchester, Manchester, UK [3]Department of Computer Science and Engineering, The Hong Kong University of Science and Technology, Hong Kong, China. Correspondence to: Liang Bai <bailiang@sxu.edu.cn>.

*Proceedings of the 43rd International Conference on Machine Learning*, Seoul, South Korea. PMLR 306, 2026. Copyright 2026 by the author(s).

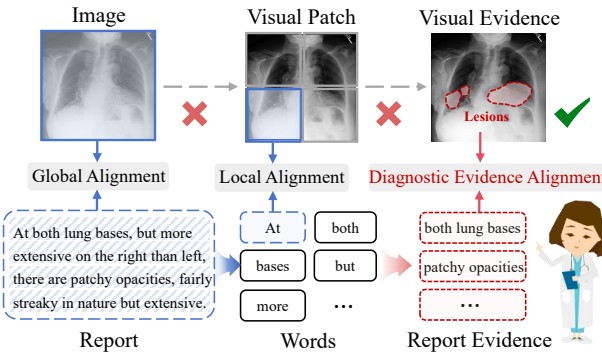

*Figure 1.* Motivation of LGDEA. Global and local alignment may overlook diagnostic evidence, whereas LGDEA aligns images and reports in a shared diagnostic evidence space.

correspondences between medical images and clinical reports (Zhu et al., 2025), which plays an important role in enabling clinically meaningful diagnostic decision-making, and is therefore widely applied to downstream tasks such as disease classification and image–report retrieval (Zou & Yin, 2025; Lu et al., 2025). Early medical VLP methods typically adapt the CLIP framework to the medical domain (Zhang et al., 2025a), employing global contrastive objectives to align entire medical images with their corresponding full clinical reports on substantial paired data, thereby learning transferable multimodal representations (Gijsen & Ritter, 2025; Li et al., 2025). This paradigm implicitly assumes that global alignment is sufficient to capture diagnostic semantics. However, radiology reports often describe multiple localized pathological findings, while diagnostically critical visual lesions are typically subtle and spatially localized. To mitigate this issue, some methods introduce local alignment by matching report words or phrases with image regions. Such local alignment methods partially alleviate the limitation of global alignment in modeling localized diagnostic cues, enabling the model to attend to image regions that are relevant to pathological descriptions (Wu et al., 2023; Li et al., 2024; Yan et al., 2025).

So far, state-of-the-art global and local alignment methods share two common characteristics: 1) They heavily rely on substantial and high-quality paired data to provide sufficient cross-modal supervisory signals. 2) They essentially

follow a feature-level alignment paradigm, which is well suited for scenarios such as natural image–text pairs where semantic correspondences are relatively explicit. However, in the medical domain, high-quality paired data are inherently limited, and diagnostic semantics are often implicit in the integrated reasoning over a small number of diagnostic evidence, which cannot be adequately captured by feature similarity alone (Xie et al., 2024; Zhang et al., 2025b). Consequently, as illustrated in Figure 1, under limited paired supervision in medical settings, global alignment methods tend to overlook fine-grained diagnostic details (Yang et al., 2024), allowing non-diagnostic information to dominate the learned embeddings, whereas local alignment methods overemphasize local correspondences, shifting the learning objective toward low-level visual details and making it difficult to acquire high-level diagnostic evidence that underpin clinical decision-making (Yang et al., 2026).

Based on these observations, we argue that effective medical VLP under limited pairing should follow the real diagnostic process of clinicians, shifting from feature-level alignment to *diagnostic evidence alignment*. However, performing diagnostic evidence alignment under limited paired data poses **two key challenges**: (1) How to construct a reliable diagnostic evidence space. (2) How to effectively learn diagnostic evidence alignment when paired data are scarce, but additional unimodal data are available without explicit cross-modal correspondences. To address these challenges, we propose **LGDEA**, an LLM-Guided Diagnostic Evidence Alignment pretraining framework that leverages medical knowledge to compensate for limited paired data. Specifically, we first utilize large language models (LLMs) to extract key diagnostic evidence from radiology reports and organize them into a structured cross-modal diagnostic evidence space. Then, report-derived evidence from limited paired data are used to guide lesion-level visual representation learning, aligning visual evidence within the diagnostic evidence space. Finally, based on evidence-centric representations, we infer cross-modal relations for unimodal data by propagating sparse paired supervision over image–image and text–text evidence graphs, thereby guiding evidence-aware cross-modal alignment over both limited paired data and abundant unpaired images and reports. Our main contributions are summarized as follows:

- We leverage LLMs to construct a cross-modal diagnostic evidence space, establishing a diagnostic evidence foundation for medical multimodal pretraining.

- We propose an evidence-aware cross-modal alignment method that effectively leverages abundant unpaired medical images and reports, thereby substantially reducing the reliance on paired data.

- Extensive experiments show that our method consistently improves performance on phrase grounding,

image–text retrieval, and zero-shot classification, even outperforming methods trained with more paired data.

## 2. Related Work

### 2.1. Global Alignment in Medical VLP

Inspired by the CLIP-style contrastive learning framework (Radford et al., 2021; Chen & Hong, 2024), numerous studies aligns medical images and their corresponding reports by projecting them into a shared embedding space. PubMedCLIP (Eslami et al., 2023) adapts CLIP to medical data to improve medical visual question answering, while MedCLIP (Wang et al., 2022b) performs alignment at the disease level using similarity matrices. CARZero (Lai et al., 2024) further enhances alignment accuracy by introducing cross-attention between image and report features to capture more complex semantic relationships. Despite these advances, existing medical vision–language pre-training methods largely rely on global representation alignment, overlooking the localized and multi-lesion nature of radiological data (Lian et al., 2025). Under limited paired supervision, such strategies often damage diagnostic signals, allowing non-diagnostic information to dominate the embedding space and weakening clinically meaningful semantics.

### 2.2. Local Alignment in Medical VLP

To address the limitations of global contrastive learning in capturing localized pathology, prior work has explored fine-grained vision–language alignment strategies (Boecking et al., 2022; Liang et al., 2025; Yu et al., 2025). GLoRIA (Huang et al., 2021) and MGCA (Wang et al., 2022a) incorporate local or multi-granularity contrastive objectives to align words or tokens with attention-weighted image regions, while MedKLIP (Wu et al., 2023) and MedAligner (Yan et al., 2025) leverage structured reports or explicit word–region alignment to enhance local semantic matching. AFLoc (Yang et al., 2026) further introduces multi-level semantic contrastive learning for unlabeled pathological localization. However, these approaches mainly focus on region-level correspondences (Chen et al., 2025), whereas medical diagnosis relies on identifying and integrating key diagnostic evidence at the sample level. Under limited paired supervision, neither global alignment nor isolated local matching is sufficient, motivating a shift toward diagnostic evidence–level alignment.

## 3. Method

### 3.1. Problem Settings

We consider a realistic medical vision–language pretraining setting where only a small subset of images and radiology reports are paired, while the majority are unpaired. Let

$\mathcal{D}_p = \{(\mathbf{I}_p, \mathbf{R}_p)\}_{p=1}^{N_p}$ denote paired samples, and $\mathcal{D}_u^I = \{\mathbf{I}_u\}_{u=1}^{N_u^I}$ and $\mathcal{D}_u^R = \{\mathbf{R}_u\}_{u=1}^{N_u^R}$ denote unpaired images and reports. We extract representations using an image encoder $f_{\text{image}}$ (Dosovitskiy et al., 2021) and a text encoder $f_{\text{text}}$ (Alsentzer et al., 2019): $f_{\text{image}} : \mathbf{I} \rightarrow (\mathbf{I}^g, \mathbf{I}^l)$, $f_{\text{text}} : \mathbf{R} \rightarrow (\mathbf{R}^g, \mathbf{R}^l)$, where $\mathbf{I}^g$ and $\mathbf{R}^g$ denote global embeddings, and $\mathbf{I}^l$ and $\mathbf{R}^l$ denote local patch- and token-level embeddings.

Existing CLIP-style pretraining methods rely on global contrastive alignment over substantial paired data, which is formulated as:

$$\mathcal{L}_g^{R \leftarrow I} = -\frac{1}{B} \sum_{p=1}^{B} \log \frac{\exp(\mathbf{I}_p^g \cdot \mathbf{R}_p^g / \tau_1)}{\sum_{k=1}^{B} \exp(\mathbf{I}_p^g \cdot \mathbf{R}_k^g / \tau_1)}, \quad (1)$$

where $B$ is the batch size and $\tau_1$ is a temperature parameter. The dot operator $\cdot$ refers to the inner product of vectors. Similarly, the loss for aligning text to images, $\mathcal{L}_g^{I \leftarrow R}$, is defined symmetrically to $\mathcal{L}_g^{R \leftarrow I}$. The global alignment loss is $\mathcal{L}_g = \mathcal{L}_g^{R \leftarrow I} + \mathcal{L}_g^{I \leftarrow R}$.

Under limited paired data, global alignment often fails to capture sparse and localized diagnostic information, while local alignment tends to overemphasize fragmented details. Therefore, effective medical vision–language pretraining should follow the actual clinical diagnostic process, shifting from feature-level alignment to diagnostic evidence alignment. However, evidence-level alignment under scarce paired supervision is non-trivial, as it requires building a reliable diagnostic evidence space and learning effective cross-modal alignment by leveraging abundant unpaired unimodal data as shown in Figure 2.

### 3.2. LLM-guided Diagnostic Evidence Space

We leverage LLMs to construct a structured cross-modal diagnostic evidence space from data with limited pairing and abundant unpaired samples, providing a diagnostic evidence foundation for medical multimodal pretraining. First, we extract diagnostic evidence from reports and organize them into a modality-shared diagnostic evidence space:

$$\mathcal{E}(\mathbf{R}) = \{e_n\}_{n=1}^{N_R} = \text{LLM}(\mathbf{R}), \quad (2)$$

where each evidence phrase $e_n$ corresponds to a concise and clinically meaningful abnormal finding. Given an evidence phrase $e_n$, we encode it into a continuous embedding using a text encoder $\mathbf{z}_n = f_{\text{text}}(e_n)$. Second, we introduce a set of learnable diagnostic prototypes:

$$\mathcal{C} = \{\boldsymbol{\mu}_k\}_{k=1}^{K}, \qquad \boldsymbol{\mu}_k \in \mathbb{R}^d. \quad (3)$$

Each prototype represents a latent diagnostic concept that may be shared across different reports and images. We compute a soft assignment over the diagnostic prototypes:

$$p(k \mid \mathbf{z}_n) = \frac{\exp(\mathbf{z}_n^\top \boldsymbol{\mu}_k / \tau_t)}{\sum_{j=1}^{K} \exp(\mathbf{z}_n^\top \boldsymbol{\mu}_j / \tau_t)}, \quad (4)$$

where $\tau_t$ is a temperature parameter. We jointly learn the text encoder and diagnostic prototypes by encouraging each evidence embedding to be reconstructed from the prototype space:

$$\mathcal{L}_{\text{rec}} = \sum_{e_n \in \mathcal{E}(R)} \left\| \mathbf{z}_n - \sum_{k=1}^{K} p(k \mid \mathbf{z}_n) \boldsymbol{\mu}_k \right\|_2^2 + \sum_{k=1}^{K} \|\boldsymbol{\mu}_k\|_2^2. \quad (5)$$

Through this mechanism, unpaired reports can effectively participate in representation learning by providing weak semantic supervision at the evidence level. This diagnostic evidence space provides a foundation for cross-modal learning of visual evidence.

Medical images often contain multiple localized pathological regions whose visual manifestations are subtle, spatially sparse, and difficult to annotate. Under limited pairing, global alignment is insufficient to guide models to focus on such diagnostically critical regions.

We therefore ground visual learning in the shared diagnostic evidence space established from text, enabling evidence-level alignment between images and reports. First, we introduce a set of $L$ learnable lesion queries $\{\mathbf{q}_\ell\}_{\ell=1}^{L}$ that attend to patch-level visual features. Given a medical image $\mathbf{I}$, a vision encoder produces patch embeddings $\mathbf{I}^l \in \mathbb{R}^{P \times d_v}$. Each lesion query interacts with the patch features through a query-based attention mechanism: $\mathbf{v}_\ell = \text{Attn}(\mathbf{q}_\ell, \mathbf{I}^l, \mathbf{I}^l)$, $\ell = 1, \ldots, L$, yielding a set of lesion-level embeddings $\mathbf{v}_\ell \in \mathbb{R}^{d_v}$. These embeddings represent latent visual evidence corresponding to potential pathological regions.

To enable evidence-level alignment with text, each lesion embedding is projected into the shared diagnostic prototype space learned from reports:

$$\mathbf{Q}_I(\ell, k) = \frac{\exp(\phi(\mathbf{v}_\ell)^\top \boldsymbol{\mu}_k / \tau_p)}{\sum_{j=1}^{K} \exp(\phi(\mathbf{v}_\ell)^\top \boldsymbol{\mu}_j / \tau_p)}, \quad (6)$$

where $\boldsymbol{\mu}_k$ denotes the $k$-th diagnostic prototype. The resulting distribution $\mathbf{Q}_I(\ell, \cdot)$ characterizes the diagnostic evidence associated with each lesion. Lesion-level prototype distributions are aggregated to obtain an image-level diagnostic distribution:

$$\bar{\mathbf{Q}}_I = \frac{1}{L} \sum_{\ell=1}^{L} \mathbf{Q}_I(\ell, \cdot). \quad (7)$$

For images paired with radiology reports, report-derived diagnostic evidence provide explicit semantic guidance. Given a report $\mathbf{R}_\mathbf{p}$ paired with an image $\mathbf{I}_\mathbf{p}$, we aggregate prototype assignments of all extracted report evidence to form a report-induced diagnostic distribution:

$$\bar{\mathbf{Q}}_R(k) = \frac{1}{|\mathcal{E}(\mathbf{R}_\mathbf{p})|} \sum_{e_n \in \mathcal{E}(\mathbf{R}_\mathbf{p})} p(k \mid \mathbf{z}_n). \quad (8)$$

**(a) LLM-Guided Diagnostic Evidence Space**

**(b) Evidence-Guided Cross-modal Alignment**

*Figure 2.* Overview of the proposed LGDEA framework. (a) LLMs extract diagnostic evidence from radiology reports, and both report evidence and lesion-level visual cues are projected into a shared diagnostic evidence space. (b) Under limited pairing, paired evidence links are used as seed edges to align report and image nodes, while report–report and image–image graphs propagate evidence-aware relations to leverage abundant unpaired reports and images for vision–language pretraining.

We then align visual evidence with textual evidence by minimizing the KL divergence:

$$\mathcal{L}_p^{evid} = \mathrm{KL}\left(\bar{\mathbf{Q}}_R \,\|\, \bar{\mathbf{Q}}_I\right), \qquad (9)$$

where the report-induced distribution $\bar{\mathbf{Q}}_R$ serves as a teacher signal. This evidence-level distillation enables paired images to learn lesion-level visual evidence consistent with diagnostic semantics, without requiring explicit lesion annotations.

For unpaired images, report-derived supervision is unavailable. However, unpaired images still exhibit meaningful visual structure and similarity patterns, which can be exploited to stabilize evidence learning. Specifically, we collect all lesion instances from both paired and unpaired images within a mini-batch as $\{\mathbf{v}_i, \mathbf{Q}_i\}_{i=1}^{N_L}$, where $N_L = B \times L$. We measure visual similarity using cosine similarity between lesion embeddings and define the $k$-nearest neighbors as $\mathcal{N}_k(i) = \mathrm{TopK}_{j \neq i}\left(\cos(\mathbf{v}_i, \mathbf{v}_j)\right)$. To encourage evidence-level consistency, we align the diagnostic prototype distributions of visually similar lesions using a similarity-weighted KL divergence:

$$\mathcal{L}_u^{evid} = \frac{1}{N_L} \sum_{i=1}^{N_L} \sum_{j \in \mathcal{N}_k(i)} w_{ij} \, \mathrm{KL}\left(\mathbf{Q}_i \,\|\, \mathrm{stopgrad}(\mathbf{Q}_j)\right), \qquad (10)$$

where $w_{ij}$ is computed by a softmax over cosine similarities. This consistency constraint allows unpaired images to actively participate in training by absorbing diagnostic semantics from nearby, evidence-aligned lesions, thereby an-

choring their lesion-level representations within the shared diagnostic evidence space.

### 3.3. Evidence-guided Cross-modal Alignment

To enable medical vision–language pretraining under limited paired data, we propose an evidence-guided weakly-supervised alignment strategy. The key challenge is that when explicit paired relations are scarce or even absent, direct image–report alignment becomes unreliable. Instead, we aim to learn higher-order cross-modal relations that reflect evidence-level semantic relatedness beyond direct pairing, and use them as weak supervision for training.

Given a mini-batch of $B$ images and $B$ reports, we define an evidence-aware cross-modal alignment loss as

$$\mathcal{L}_{\text{evi-align}} = \mathcal{L}_{\text{evi}}^{R \leftarrow I} + \mathcal{L}_{\text{evi}}^{I \leftarrow R}, \qquad (11)$$

where the image-to-report direction is

$$\mathcal{L}_{\text{evi}}^{R \leftarrow I} = -\frac{1}{B} \sum_{i=1}^{B} \sum_{j=1}^{B} \mathbf{P}_{ij} \log \frac{\exp\left(\mathbf{H}_{\mathbf{I}_i}^\top \mathbf{H}_{\mathbf{R}_j}/\tau_2\right)}{\sum_{k=1}^{B} \exp\left(\mathbf{H}_{\mathbf{I}_i}^\top \mathbf{H}_{\mathbf{R}_k}/\tau_2\right)}, \qquad (12)$$

and $\mathcal{L}_{\text{evi}}^{I \leftarrow R}$ is defined symmetrically. Here, $\tau_2$ is the temperature hyperparameter, and $\mathbf{P}_{ij} \in [0, 1]$ denotes the higher-order evidence relation between image $\mathbf{I}_i$ and report $\mathbf{R}_j$. Unlike hard paired labels, $\mathbf{P}_{ij}$ provides soft supervision by measuring how likely two samples share similar diagnostic evidence.

For each report $\mathbf{R}$, we extract a set of diagnostic evidence

$\mathcal{E}(\mathbf{R})$ using an LLM, and obtain its report-level diagnostic representation by aggregating evidence embeddings:

$$\mathbf{H_R} = \frac{1}{|\mathcal{E}(\mathbf{R})|} \sum_{e_n \in \mathcal{E}(\mathbf{R})} \mathbf{z}_n \in \mathbb{R}^d. \qquad (13)$$

Similarly, for an image $\mathbf{I}$ with $L$ lesion queries, we aggregate lesion-level visual evidence projected into the diagnostic evidence embedding space:

$$\mathbf{H_I} = \frac{1}{L} \sum_{\ell=1}^{L} \phi(\mathbf{v}_\ell) \in \mathbb{R}^d, \qquad (14)$$

where $\phi(\cdot)$ maps lesion embeddings $\mathbf{v}_\ell$ into the diagnostic evidence embedding space.

Let $\mathbf{Y} \in \{0,1\}^{N_I \times N_R}$ denote the sparse binary relation matrix induced by paired image–report samples, where $\mathbf{Y}_{ij} = 1$ indicates that $\mathbf{I}_i$ and $\mathbf{R}_j$ form a paired instance. When paired supervision is available, we can directly set $\mathbf{P} = \mathbf{Y}$, leading to evidence-aware contrastive learning with hard positives.

However, under scarce paired supervision, $\mathbf{Y}$ is extremely sparse and cannot provide sufficient alignment signals. To address this limitation, we infer higher-order relations by propagating the sparse seed relations over intra-modal evidence graphs, which effectively transfers supervision from paired data to unpaired unimodal data.

We first construct a report–report evidence graph based on report-level representations:

$$A_{TT}(i,j) = \mathrm{sim}(\mathbf{H_{R_i}}, \mathbf{H_{R_j}}), \qquad (15)$$

and row-normalize it to obtain the text-side propagation matrix $S_T$. Likewise, we construct an image–image evidence graph:

$$A_{II}(i,j) = \mathrm{sim}(\mathbf{H_{I_i}}, \mathbf{H_{I_j}}), \qquad (16)$$

yielding the image-side propagation matrix $S_I$ after normalization.

Starting from $\mathbf{P}^{(0)} = \mathbf{Y}$, we perform label propagation to infer higher-order cross-modal relations:

$$\mathbf{P}^{(t+1)} = S_I \mathbf{P}^{(t)} S_T + \mathbf{Y}, \qquad (17)$$

where the residual term $\mathbf{Y}$ preserves original paired relations and stabilizes propagation. After two propagation steps, we obtain higher-order relations $\mathbf{P}$ and apply row-wise normalization. The resulting $\mathbf{P}$ enables us to treat evidence-similar image–report pairs as weak positives, thereby supporting evidence-aware cross-modal alignment even when explicit pairing is limited as shown in Figure 3.

The overall training objective integrates diagnostic evidence modeling and cross-modal alignment:

$$\mathcal{L} = \mathcal{L}_{\mathrm{rec}} + \mathcal{L}_p^{\mathrm{evid}} + \mathcal{L}_u^{\mathrm{evid}} + \mathcal{L}_{\mathrm{evi\text{-}align}}, \qquad (18)$$

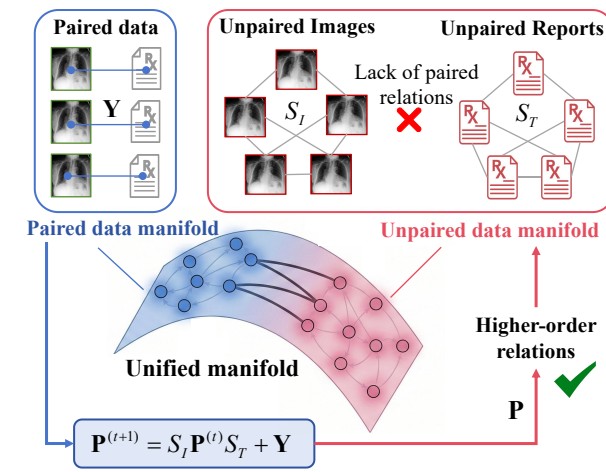

*Figure 3.* Evidence-guided higher-order alignment under scarce paired supervision. Sparse paired links $\mathbf{Y}$ are grounded in a shared diagnostic evidence space and propagated over intra-modal evidence graphs to infer higher-order relations $\mathbf{P}$.

where $\mathcal{L}_{\mathrm{rec}}$ learns a structured diagnostic evidence space from reports, $\mathcal{L}_p^{\mathrm{evid}}$ aligns the limited paired samples within this diagnostic evidence space, $\mathcal{L}_u^{\mathrm{evid}}$ enforces evidence consistency for unpaired images, and $\mathcal{L}_{\mathrm{evi\text{-}align}}$ performs evidence-guided cross-modal alignment. A detailed algorithm description is in the appendix A.5.

## 4. Experiment

### 4.1. Experimental Settings

We evaluate LGDEA under both single-domain and cross-domain medical vision–language pretraining settings. Below we briefly describe the training data, downstream evaluation benchmarks, and baselines. Additional implementation details are provided in the Appendix A.

**Training Data.** We conduct pretraining primarily on the MIMIC-CXR dataset (Johnson et al., 2019), which contains substantial chest X-ray images and corresponding radiology reports. To simulate realistic scenarios with limited pairing, we randomly sample different proportions of paired image–report data, while treating the remaining images and reports as unpaired unimodal data. For cross-domain pretraining, we additionally incorporate images from CheXpert (Irvin et al., 2019) as unpaired visual data.

**Downstream Tasks.** We evaluate learned representations on a diverse set of downstream medical vision–language tasks, including phrase grounding, image–text retrieval, and zero-shot classification. Specifically, we use MS-CXR for phrase grounding, MIMIC-5×200 for retrieval and zero-shot classification, and RSNA Pneumonia, COVID, and NIH Chest X-rays for disease classification.

*Table 1.* The performance of the Phrase Grounding task is evaluated on the MS-CXR dataset using Contrast-to-Noise Ratio (CNR) scores across eight disease categories.

| Group | Model | Atelectasis | Cardiomegaly | Consolidation | Lung Opacity | Edema | Pneumonia | Pneumothorax | Pleural Effusion |
|---|---|---|---|---|---|---|---|---|---|
| Paired Baseline | MedCLIP | 0.5554 | 0.5253 | 0.5767 | 0.5599 | 0.4433 | 0.6175 | 0.4686 | 0.5076 |
| | GLoRIA | 0.6088 | 0.5253 | 0.6194 | 0.5376 | 0.5062 | 0.7308 | 0.5896 | 0.8991 |
| | BioViL | 0.5376 | 0.5014 | 0.5852 | 0.5266 | 0.4727 | 0.4602 | 0.4882 | 0.5841 |
| | MGCA | 0.6338 | 0.5680 | 0.6543 | 0.5586 | 0.5247 | 0.8324 | 0.5240 | 0.8207 |
| | MedKLIP | 0.5570 | 0.5282 | 0.5921 | 0.5926 | 0.4905 | 0.6781 | 0.5858 | 0.6398 |
| | PRIOR | 0.6452 | 0.5601 | 0.5876 | 0.6624 | 0.5548 | 0.6693 | 0.6564 | 0.8221 |
| | CLEFT | 0.6871 | 0.5687 | 0.6492 | 0.5699 | 0.4989 | 0.7259 | 0.5976 | 0.7974 |
| | MAVL | 0.5811 | 0.5417 | 0.5486 | 0.6381 | 0.5115 | 0.7056 | 0.5952 | 0.7601 |
| | CARZero | 0.6757 | 0.5214 | 0.6889 | 0.5572 | 0.5209 | 0.8438 | 0.6326 | 0.9672 |
| | AFLoc | 0.7941 | 0.5928 | 0.7688 | 0.6894 | 0.6271 | 0.7484 | 0.6810 | 0.9866 |
| | MedAligner | 0.8312 | 0.6058 | 0.8144 | 0.7163 | 0.7335 | 0.8489 | 0.7249 | 1.0207 |
| Single Domain | LGDEA$_{5\%pair}$ | 0.8000 | 0.9127 | 0.8071 | 0.9032 | 0.8081 | 0.7849 | 0.7902 | 0.9742 |
| | LGDEA$_{10\%pair}$ | **0.8449** | **0.9525** | **0.8357** | **0.9564** | **0.8172** | **0.9353** | **0.8526** | **1.0928** |
| Cross Domain | LGDEA$_{5\%pair}$ | 0.8011 | 0.9111 | 0.7975 | 0.8789 | 0.7868 | 0.8235 | 0.7067 | 0.6667 |
| | LGDEA$_{10\%pair}$ | 0.8306 | 0.9416 | 0.8126 | 0.9003 | 0.7891 | 0.9214 | 0.7952 | 0.7519 |

**Baselines.** We compare LGDEA with representative state-of-the-art medical VLP methods, including Med-CLIP (Wang et al., 2022b), MGCA (Wang et al., 2022a), MedKLIP (Wu et al., 2023), CLEFT (Du et al., 2024), MAVL (Phan et al., 2024), PRIOR (Cheng et al., 2023), CARZero (Lai et al., 2024), MedAligner (Yan et al., 2025), and AFLoc (Yang et al., 2026). All baselines are trained under consistent data settings for fair comparison.

### 4.2. Experimental Results

We evaluate the proposed LGDEA framework under both single-domain and cross-domain medical vision–language pretraining settings with limited paired image–report. In the single-domain setting, all data are drawn from MIMIC-CXR, where only a small proportion (5% or 10%) of image–report pairs is retained and the remaining images and reports are included without explicit cross-modal pairing. In the cross-domain setting, paired samples and reports are from MIMIC-CXR, while images from CheXpert are incorporated as an additional visual source. Unimodal reports are derived from the remaining MIMIC-CXR reports. Across different pairing ratios, LGDEA is evaluated on multiple downstream tasks, including phrase grounding, image–text retrieval and zero-shot classification, assessing fine-grained diagnostic localization, cross-modal semantic alignment, and generalization to unseen diseases and domains.

**Phrase Grounding**. Phrase grounding aims to accurately associate textual phrases with their corresponding regions in medical images, thereby enabling fine-grained localization of diagnostic cues and enhancing model interpretabil-

| Phrases | Image | LGDEA | GT |
|---|---|---|---|
| small right apical pneumothorax | | | |
| enlarged cardiac silhouette | | | |

*Figure 4.* Attention maps for two disease categories are visualized on the MS-CXR dataset, comparing LGDEA with GT. Red bounding boxes indicate the ground truth regions relevant to phrase grounding. Highlighted pixels correspond to higher activation weights, reflecting stronger associations between specific diagnostic terms and image regions.

ity. Table 1 summarizes the phrase grounding results on the MS-CXR dataset, evaluated using the contrast-to-noise ratio (CNR) (Hendrick, 2008). With only 10% of the paired image–text data, LGDEA achieves the highest CNR scores across all eight disease categories, substantially outperforming methods that leverage fully paired image–text relations and focus on local alignment modeling, such as AFLoc and MedAligner. Even when trained on cross-domain image–text data, LGDEA achieves higher CNR scores across all eight disease categories using only 10% of the paired image–text samples. In addition, visualization results based on Grad-CAM (Selvaraju et al., 2017) heatmaps (Figure 4) further demonstrate that LGDEA can accurately localize

*Table 2.* The performance of baselines trained with limited paired image–text data on the phrase grounding task on the MS-CXR dataset, evaluated using the contrast-to-noise ratio (CNR) score across eight disease categories.

| Group | Model | Atelectasis | Cardiomegaly | Consolidation | Lung Opacity | Edema | Pneumonia | Pneumothorax | Pleural Effusion |
|---|---|---|---|---|---|---|---|---|---|
| Single Domain | GLoRIA$_{50\%\text{pair}}$ | 0.4486 | 0.3050 | 0.4359 | 0.4479 | 0.2069 | 0.4941 | 0.3017 | 0.3533 |
| | MGCA$_{50\%\text{pair}}$ | 0.4620 | 0.2966 | 0.4363 | 0.4897 | 0.2352 | 0.6071 | 0.3569 | 0.4338 |
| | MedAligner$_{50\%\text{pair}}$ | 0.6346 | 0.4149 | 0.5344 | 0.5431 | 0.5889 | 0.6836 | 0.6021 | 0.7298 |
| | LGDEA$_{5\%\text{pair}}$ | 0.8000 | 0.9127 | 0.8071 | 0.9032 | 0.8081 | 0.7849 | 0.7902 | 0.9742 |
| | LGDEA$_{10\%\text{pair}}$ | **0.8449** | **0.9525** | **0.8357** | **0.9564** | **0.8172** | **0.9353** | **0.8526** | **1.0928** |
| Cross Domain | LGDEA$_{5\%\text{pair}}$ | 0.8011 | 0.9111 | 0.7975 | 0.8789 | 0.7868 | 0.8235 | 0.7067 | 0.6667 |
| | LGDEA$_{10\%\text{pair}}$ | 0.8306 | 0.9416 | 0.8126 | 0.9003 | 0.7891 | 0.9214 | 0.7952 | 0.7519 |

disease-related phrases to the corresponding pathological regions. More visualizations are in the appendix B.1.

To ensure a fair comparison under limited-pairing settings, we retrain paired-supervision-based baselines by reducing the proportion of available paired image–report data. The phrase grounding results on MS-CXR are reported in Table 2. As shown, these baselines suffer clear performance drops when paired supervision is reduced. Even with 50% paired data, they still obtain lower CNR scores than LGDEA trained with only 5% or 10% paired samples. This demonstrates the advantage of LGDEA in effectively leveraging limited paired data and abundant unpaired unimodal data.

**Image-Text Retrieval.** To evaluate the matching performance between medical images and textual descriptions, we conducted experiments on the MIMIC-5 × 200 dataset. Specifically, for each query image, we computed the cosine similarity between its [CLS] token representation and the representations of candidate sentences for retrieval. Retrieval performance was assessed using the Precision@K metric. The results are summarized in Table 4. Although PRIOR and CARZero rely on 100% paired image–text data and benefit from their global image–text alignment to achieve competitive performance, LGDEA, using only 10% of the paired image–text relations, surpasses them on Precision@1, Precision@2, Precision@5, and Precision@10. Moreover, with only 10% of the paired image–text relations, LGDEA outperforms the best-performing MedAligner in terms of Precision@1 and Precision@10, while achieving comparable precision on Precision@2 and Precision@5. To better illustrate LGDEA semantic understanding capabilities, we present a case study of image-text retrieval examples in Figure 5.

**Hard-negative Image–Text Retrieval.** To further evaluate whether LGDEA captures fine-grained clinically relevant semantics, we conduct a hard-negative image–text retrieval experiment on MIMIC-5×200. For each image, the model is required to distinguish the ground-truth report from four semantically perturbed alternatives, including negation, uncertainty, anatomy, and comparison perturbations. Top-

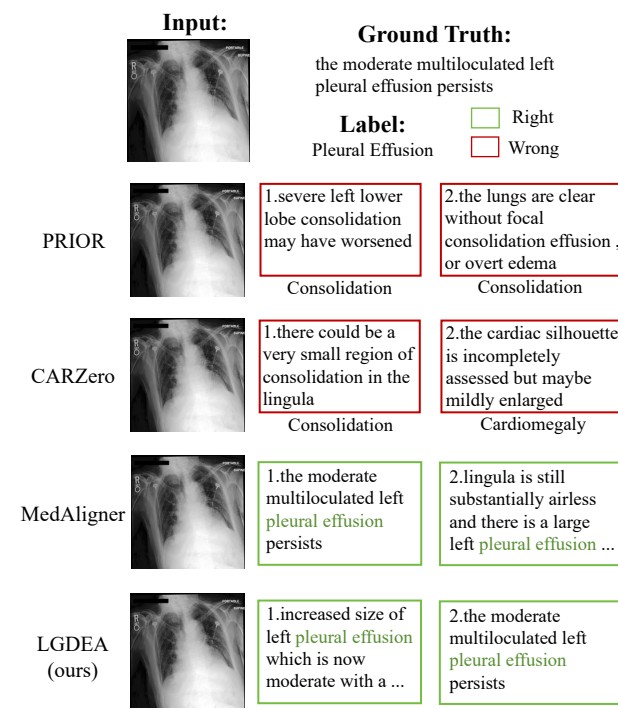

*Figure 5.* Qualitative results of image-to-text retrieval. We show the top two retrieved texts for LGDEA, with comparisons to three baselines. We note the categories below wrongly retrieved samples.

1 ACC denotes the proportion of samples for which the ground-truth report is ranked first, while MRR measures the average ranking quality of the ground-truth report. As shown in Table 3, LGDEA achieves an average Top-1 ACC of 63.75%, substantially higher than the 20% accuracy expected under random guessing among five candidates. The average MRR reaches 0.7454, indicating that the ground-truth report is generally ranked near the top. These results suggest that LGDEA is effective in discriminating subtle clinical semantic differences, supporting its ability to learn diagnostic evidence beyond coarse image–text matching.

**Zero-Shot Classification.** To assess the adaptability and generalization capability of the learned image encoder, we

*Table 3.* Hard-negative image–text retrieval on MIMIC-5×200. For each image, the model ranks the ground-truth report against four semantically perturbed alternatives. All metrics are reported in accuracy (%). Top-1 ACC, equivalent to "Pred. as Positive", denotes the proportion of samples for which the ground-truth report is ranked first. MRR (%) measures the ranking quality of the ground-truth report. The remaining "Pred. as" columns indicate which type of hard-negative candidate is incorrectly ranked first.

| Disease | Top-1 ACC | MRR | Pred. as Positive | Pred. as Negation | Pred. as Uncertainty | Pred. as Anatomy | Pred. as Comparison |
|---|---|---|---|---|---|---|---|
| Atelectasis | 56.50 | 69.47 | 56.50 | 26.00 | 0.00 | 17.00 | 0.50 |
| Cardiomegaly | 61.42 | 72.71 | 61.42 | 29.95 | 0.51 | 7.61 | 0.51 |
| Consolidation | 69.35 | 79.53 | 69.35 | 21.11 | 0.00 | 9.05 | 0.50 |
| Edema | 68.00 | 76.95 | 68.00 | 26.50 | 0.00 | 5.50 | 0.00 |
| Pleural Effusion | 63.50 | 74.03 | 63.50 | 27.50 | 0.00 | 9.00 | 0.00 |
| **Average** | **63.75** | **74.54** | **63.75** | 26.21 | 0.10 | 9.63 | 0.30 |

*Table 4.* The Image–Text Retrieval task is compared with state-of-the-art methods on the MIMIC-5 × 200 dataset, and Precision (%) scores are reported.

| Group | Model | Prec@1 | Prec@2 | Prec@5 | Prec@10 |
|---|---|---|---|---|---|
| Paired Baseline | MedCLIP | 44.40 | 43.45 | 44.50 | 45.58 |
| | GLoRIA | 42.38 | 45.65 | 47.74 | 41.98 |
| | BioViL | 48.81 | 47.88 | 50.05 | 32.88 |
| | MGCA | 50.06 | 49.77 | 49.05 | 47.53 |
| | MedKLIP | 50.00 | 50.00 | 49.42 | 50.00 |
| | PRIOR | 53.19 | 51.72 | 51.89 | 41.69 |
| | CLEFT | 52.75 | 50.31 | 48.75 | 49.81 |
| | MAVL | 50.00 | 50.62 | 51.06 | 49.97 |
| | CARZero | 50.00 | 50.00 | 51.65 | 47.38 |
| | AFLoc | 54.37 | 52.78 | 49.79 | 43.20 |
| | MedAligner | 55.88 | **54.08** | **54.05** | 50.56 |
| Single Domain | LGDEA$_{5\%pair}$ | 53.25 | 51.03 | 50.66 | 49.98 |
| | LGDEA$_{10\%pair}$ | **56.31** | 53.85 | 53.71 | **50.65** |
| Cross Domain | LGDEA$_{5\%pair}$ | 50.88 | 50.68 | 48.57 | 48.33 |
| | LGDEA$_{10\%pair}$ | 53.58 | 51.23 | 50.06 | 49.14 |

*Table 5.* The accuracy (%) of the Zero-shot Classification is evaluated on the MIMIC-5 × 200, COVID, RSNA Pneumonia, and NIH Chest X-rays datasets.

| Group | Model | MIMIC-5 × 200 | COVID | RSNA | NIH Chest X-rays |
|---|---|---|---|---|---|
| Paired Baseline | MedCLIP | 52.50 | 77.70 | 79.90 | 58.84 |
| | GLoRIA | 72.22 | 87.77 | 78.55 | 59.47 |
| | BioViL | 73.38 | 80.10 | 78.46 | 61.33 |
| | MGCA | 74.87 | 86.87 | 79.20 | 68.40 |
| | MedKLIP | 51.94 | 83.26 | 74.65 | 79.40 |
| | PRIOR | 76.83 | 86.27 | 74.73 | 80.84 |
| | CLEFT | 75.47 | 84.18 | 78.25 | 78.95 |
| | MAVL | 74.60 | 83.06 | 78.14 | 82.77 |
| | CARZero | 76.06 | 86.80 | 78.55 | 83.16 |
| | AFLoc | 76.20 | 87.80 | 73.52 | 83.87 |
| | MedAligner | 77.89 | 88.20 | **80.05** | 84.11 |
| Single Domain | LGDEA$_{5\%pair}$ | 79.44 | 87.53 | 78.48 | 85.88 |
| | LGDEA$_{10\%pair}$ | **80.00** | **90.47** | 79.26 | **87.06** |
| Cross Domain | LGDEA$_{5\%pair}$ | 74.10 | 88.77 | 77.03 | 77.06 |
| | LGDEA$_{10\%pair}$ | 76.78 | 90.13 | 78.92 | 78.56 |

further evaluated its performance on zero-shot classification tasks. On the MIMIC-5 × 200 dataset, we perform classification evaluation by computing the similarity between the labels predicted by the image encoder and the ground-truth labels. For the COVID, RSNA Pneumonia, and NIH ChestX-ray datasets, we attach a classification head to the image encoder and train it using the cross-entropy loss, enabling the model to make predictions solely based on image features. Table 5 summarizes the classification results on MIMIC-5 × 200, COVID, RSNA Pneumonia, and NIH ChestX-ray. When trained with only 10% paired image–text data, LGDEA outperforms all baselines across most datasets, demonstrating strong cross-dataset generalization. While it performs slightly worse than MedAligner and MedCLIP on RSNA, it surpasses them on the remaining datasets.

### 4.3. Ablation Study

**The contributions of different components.** We conduct ablation studies to assess the contribution of each core component in LGDEA. All variants are evaluated on the same downstream benchmarks for fair comparison. Specifically,

we consider: (1) w/o $\mathcal{L}_{rec}$, removing diagnostic evidence prototype learning and using global text embeddings instead; (2) w/o $\mathcal{L}_p^{evid}$, removing paired evidence distillation; (3) w/o $\mathcal{L}_u^{evid}$, removing unpaired evidence consistency; and (4) w/o $\mathcal{L}_{evi\text{-}align}$, disabling evidence relation inference and using only paired samples as positives. As shown in Table 10 and 11 in the appendix B.2, removing any component degrades performance, with w/o $\mathcal{L}_{rec}$ causing the largest drop, highlighting the importance of diagnostic evidence prototype learning under limited paired supervision.

**The effects of different LLMs.** In our main experiments, diagnostic evidence is extracted using Spark-Desk. To evaluate the robustness of LGDEA to the choice of evidence extractor, we replace Spark with two open-source LLMs, Qwen-7B (Bai et al., 2023) and LLaMA-8B (Grattafiori et al., 2024), while keeping all other components unchanged. As shown in Table 6 and Table 12 in the appendix B.2, LGDEA achieves stable performance across different LLM settings on both image classification and phrase grounding tasks. Although the extracted diagnostic evidence varies

*Table 6.* The accuracy (%) of the Zero-shot Classification is evaluated on the MIMIC-5 × 200, COVID, RSNA Pneumonia, and NIH Chest X-rays datasets and compared with LLaMA and Qwen.

| Model | MIMIC-5 × 200 | COVID | RSNA | NIH Chest X-rays |
|---|---|---|---|---|
| LLaMA | 79.72 | 90.10 | **79.60** | 85.80 |
| Qwen | 79.92 | 90.39 | 79.15 | 86.14 |
| Ours | **80.00** | **90.47** | 79.26 | **87.06** |

in linguistic style across LLMs, the overall performance remains largely consistent, indicating that LGDEA does not rely on a specific LLM but instead benefits primarily from the semantic effectiveness of the diagnostic evidence.

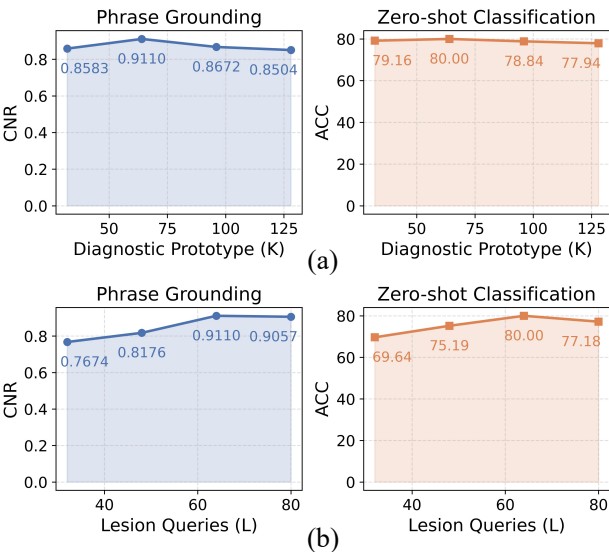

*Figure 6.* (a) Effect of the number of diagnostic prototypes $K$ and (b) effect of the number of lesion queries $L$, evaluated using average CNR on the MS-CXR phrase grounding task and ACC on the MIMIC-5 × 200 zero-shot classification task.

### 4.4. Hyper-parameter Analysis

We analyze the sensitivity of LGDEA to two key hyper-parameters: the number of diagnostic prototypes $K$, which controls the semantic capacity of the shared evidence space, and the number of lesion queries $L$, which determines the granularity of lesion-level visual modeling.

**Number of diagnostic prototypes** $K$. We vary $K \in \{32, 64, 96, 128\}$ while fixing $L = 64$. As shown in Figure 6 and Table 7, a small $K$ limits semantic expressiveness by forcing heterogeneous evidence to share prototypes, degrading phrase grounding and zero-shot classification performance. Increasing $K$ improves fine-grained evidence–lesion alignment, while overly large values lead to performance saturation and slight degradation. Overall,

$K = 64$ provides the best trade-off between expressiveness and stability and is used in all experiments.

*Table 7.* Image classification accuracy (%) on the COVID, RSNA Pneumonia, and NIH Chest X-rays datasets with varying numbers of diagnostic prototypes $K$.

| Diagnostic prototypes | COVID | RSNA | NIH Chest X-rays |
|---|---|---|---|
| $K = 32$ | 87.14 | 77.39 | 85.91 |
| $K = 64$ | **90.47** | 79.26 | **87.06** |
| $K = 96$ | 86.80 | **79.37** | 86.88 |
| $K = 128$ | 84.72 | 78.21 | 86.59 |

**Number of lesion queries** $L$. We fix $K = 64$ and vary $L \in \{32, 48, 64, 80\}$. As shown in Figure 6 and Table 8, increasing $L$ enhances coverage of multiple pathological regions and improves grounding accuracy. However, excessively large $L$ introduces redundant and overlapping queries, leading to slightly degraded performance. We therefore adopt $L = 64$ as a balanced setting across all downstream tasks.

*Table 8.* Image classification accuracy (%) on the COVID, RSNA Pneumonia, and NIH Chest X-rays datasets with different numbers of lesion queries $L$.

| Lesion queries | COVID | RSNA | NIH Chest X-rays |
|---|---|---|---|
| $L = 32$ | 85.10 | 77.71 | 85.06 |
| $L = 48$ | 86.73 | 78.95 | 85.53 |
| $L = 64$ | **90.47** | **79.26** | **87.06** |
| $L = 80$ | 85.57 | 79.16 | 86.72 |

## 5. Conclusion

In this paper, we study medical vision–language pretraining under limited paired image–report supervision. We argue that conventional global and local alignment methods may be insufficient in this setting, as they either overlook diagnostic evidence or overemphasize fragmented local correspondences. To address this issue, we propose LGDEA, which shifts pretraining from feature-level alignment to evidence-level semantic alignment. LGDEA extracts key diagnostic evidence from radiology reports, constructs a shared diagnostic evidence space, and guides lesion-level visual evidence learning within this space. It further leverages evidence-aware relations to exploit abundant unpaired images and reports. Extensive experiments on phrase grounding, image–text retrieval, and zero-shot classification demonstrate that LGDEA achieves consistent improvements and can rival methods pretrained with substantially more paired data. These results highlight the effectiveness of diagnostic evidence alignment for medical vision–language pretraining under limited pairing.

## Acknowledgements

This work is supported by the National Natural Science Foundation of China (62432006, 62276159, 62522216, 62402408), the Fundamental Research Program of Shanxi Province (202303021223004), Hong Kong SAR Research Grants Council Early Career Scheme (26208924), and Hong Kong SAR Research Grants Council General Research Fund (16219025). We thank the anonymous reviewers for their valuable and constructive comments.

## Impact Statement

This paper presents work whose goal is to advance the field of Machine Learning. There are many potential societal consequences of our work, none which we feel must be specifically highlighted here.

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

# A. Additional Experimental Details

## A.1. Training Datasets

*Table 9.* The usage of the dataset employed by the model in pre-training and downstream medical tasks.

| Dataset | Size/Images | Annotations | Pre-training | Phrase Grounding | Image-Text Retrieval | Zero-shot Classification |
|---|---|---|---|---|---|---|
| MIMIC-CXR | 377,110 | Paired images and reports | ✓ | | | |
| CheXpert | 224,316 | Images | ✓ | | | |
| MS-CXR | 206 | Bounding boxes, radiology phrase | | ✓ | | |
| MIMIC-$5 \times 200$ | 1,000 | Radiology text, 5 diseases | | | ✓ | ✓ |
| COVID | 5,162 | COVID-19/normal labels | | | | ✓ |
| RSNA Pneumonia | 12,024 | Pneumonia/normal labels | | | | ✓ |
| NIH Chest X-rays | 112,120 | 14 disease/normal labels | | | | ✓ |

**MIMIC-CXR.** MIMIC-CXR contains 377,110 chest X-ray images and 227,835 radiology reports from 65,379 patients (Johnson et al., 2019). We use the *Findings* and *Impression* sections as textual input. Following MedCLIP (Wang et al., 2022b), the dataset is split into a pretraining set and an evaluation subset (MIMIC-5×200). To simulate limited pairing, we randomly sample 5% and 10% of paired image–report data, while treating the remaining images and reports as unpaired unimodal data. The dataset is publicly available at `https://physionet.org/content/mimic-cxr-jpg/2.1.0/`.

**CheXpert.** CheXpert contains 224,316 chest radiographs from 65,240 patients (Irvin et al., 2019). In cross-domain experiments, CheXpert images are used as additional unpaired visual data, combined with paired and unpaired text data from MIMIC-CXR. The dataset is available at `https://www.kaggle.com/datasets/ashery/chexpert`.

## A.2. Downstream Evaluation Datasets

**MS-CXR.** MS-CXR is a phrase grounding benchmark with radiologist-annotated bounding boxes for eight disease categories (Boecking et al., 2022).

**MIMIC-$5\times200$.** A balanced subset constructed from MIMIC-CXR for zero-shot classification and image–text retrieval, covering five disease categories following GLoRIA (Huang et al., 2021).

**RSNA Pneumonia.** A binary chest X-ray dataset annotated for pneumonia detection with bounding boxes (Shih et al., 2019). We follow MedCLIP (Wang et al., 2022b) to construct class-balanced training and testing splits.

**COVID and NIH Chest X-rays.** We evaluate on the COVID-19 dataset (Rahman et al., 2021) and the NIH Chest X-rays dataset (Wang et al., 2017) for disease classification using official splits.

## A.3. Baselines

We compare LGDEA with MedCLIP (Wang et al., 2022b), MGCA (Wang et al., 2022a), MedKLIP (Wu et al., 2023), CLEFT (Du et al., 2024), MAVL (Phan et al., 2024), PRIOR (Cheng et al., 2023), CARZero (Lai et al., 2024), MedAligner (Yan et al., 2025), and AFLoc (Yang et al., 2026). For methods originally pretrained on CheXpert, we retrain them on MIMIC-CXR to ensure fair comparison.

## A.4. Implementation Details

LGDEA is trained with a multi-phase optimization schedule. The initial training phase uses a batch size of 128 and a learning rate of $5 \times 10^{-5}$ for 2 epochs, followed by subsequent phases trained for 5 epochs each with a batch size of 64 and a learning rate of $1 \times 10^{-4}$. All experiments use AdamW with a weight decay of $1 \times 10^{-6}$ and are conducted on two NVIDIA A100 GPUs (40GB). Images are resized to 256×256 and randomly cropped to 224×224 during training.

## A.5. Algorithm Description

---

**Algorithm 1** LGDEA: LLM-Guided Diagnostic Evidence Alignment

---

**Require:** Paired data $\mathcal{D}_p$, unpaired images $\mathcal{D}_u^I$, unpaired reports $\mathcal{D}_u^R$.
**Require:** Encoders $f_{\text{image}}, f_{\text{text}}$, prototypes $\{\mu_k\}_{k=1}^K$, lesion queries $\{q_\ell\}_{\ell=1}^L$.
**Ensure:** Trained model parameters.
1: **while** not converged **do**
2:      Sample a mini-batch of paired and unpaired images/reports.
3:      **(1) Build diagnostic evidence space.**
4:      Extract evidence $\mathcal{E}(R) = \text{LLM}(R)$ and encode $\{z_n\}$.
5:      Update evidence prototypes by minimizing $\mathcal{L}_{\text{rec}}$.
6:      **(2) Learn lesion-level visual evidence.**
7:      Obtain lesion embeddings $\{v_\ell\}$ using lesion queries and attention.
8:      Project lesions into prototype space to get $\bar{Q}_I$.
9:      For paired samples, align with report evidence via $\mathcal{L}_p^{\text{evid}}$.
10:     For unpaired images, enforce lesion consistency via $\mathcal{L}_u^{\text{evid}}$.
11:     **(3) Infer higher-order cross-modal relations.**
12:     Compute image/report evidence representations $\{H_I\}, \{H_R\}$.
13:     Construct evidence graphs $S_I, S_T$ and propagate seed links: $\mathbf{P}^{(t+1)} = S_I \mathbf{P}^{(t)} S_T + \mathbf{Y}$.
14:     **(4) Evidence-guided weakly-supervised alignment.**
15:     Optimize $\mathcal{L}_{\text{evi-align}}$ using $\mathbf{P}$-weighted contrastive loss.
16:     Update parameters by minimizing $\mathcal{L} = \mathcal{L}_{\text{rec}} + \mathcal{L}_p^{\text{evid}} + \mathcal{L}_u^{\text{evid}} + \mathcal{L}_{\text{evi-align}}$.
17: **end while**

---

## B. Additional experimental results.

### B.1. Additional Visualization Results

To further evaluate the localization ability of LGDEA, we provide additional qualitative results on the MS-CXR phrase grounding benchmark. As shown in Figure 7, LGDEA produces more concentrated attention responses around the ground-truth pathological regions across multiple disease categories. Compared with baseline methods, which may highlight irrelevant anatomical areas or generate diffuse activation maps, LGDEA better aligns disease-related textual phrases with clinically meaningful visual evidence. These results further support the effectiveness of diagnostic evidence alignment in guiding the model to focus on lesion-relevant regions under limited paired supervision.

### B.2. Ablation experimental results.

We perform an ablation study following the four variants introduced in Sec. 4.3 and evaluate all variants on phrase grounding and image–text retrieval. As shown in Tables 10 and 11, removing any component consistently degrades performance, confirming that each module contributes to the overall effectiveness of LGDEA. Among these variants, removing $\mathcal{L}_{\text{rec}}$ leads to the most significant performance drop. This indicates that the structured diagnostic evidence prototype space plays a central role in organizing report-derived evidence and providing a semantic anchor for both textual and visual evidence modeling. Without this prototype space, the model is less able to capture clinically meaningful evidence concepts, which weakens lesion-level visual learning and cross-modal semantic alignment under limited paired supervision. These results further demonstrate that diagnostic evidence modeling is essential for enabling LGDEA to effectively exploit scarce paired data and abundant unpaired unimodal data.

*Table 10.* The performance of the phrase grounding task is compared with the all ablated variants, using Contrast-to-Noise Ratio (CNR) scores across eight disease categories.

| Model | Atelectasis | Cardiomegaly | Consolidation | Lung Opacity | Edema | Pneumonia | Pneumothorax | Pleural Effusion |
|---|---|---|---|---|---|---|---|---|
| Full LGDEA | 0.8449 | 0.9525 | 0.8357 | 0.9564 | 0.8172 | 0.9353 | 0.8526 | 1.0928 |
| w/o $\mathcal{L}_{\text{rec}}$ | 0.6516 | 0.7170 | 0.6172 | 0.7604 | 0.7236 | 0.8142 | 0.6407 | 0.8125 |
| w/o $\mathcal{L}_p^{\text{evid}}$ | 0.7214 | 0.8121 | 0.7210 | 0.8946 | 0.7571 | 0.8856 | 0.7179 | 0.9072 |
| w/o $\mathcal{L}_u^{\text{evid}}$ | 0.7064 | 0.7977 | 0.6972 | 0.8828 | 0.7724 | 0.8153 | 0.7348 | 0.8524 |
| w/o $\mathcal{L}_{\text{evi-align}}$ | 0.7355 | 0.8302 | 0.8049 | 0.9150 | 0.7751 | 0.8973 | 0.6238 | 0.9196 |

In our main experiments, diagnostic evidence is extracted using Spark-Desk. To assess the robustness of LGDEA to the

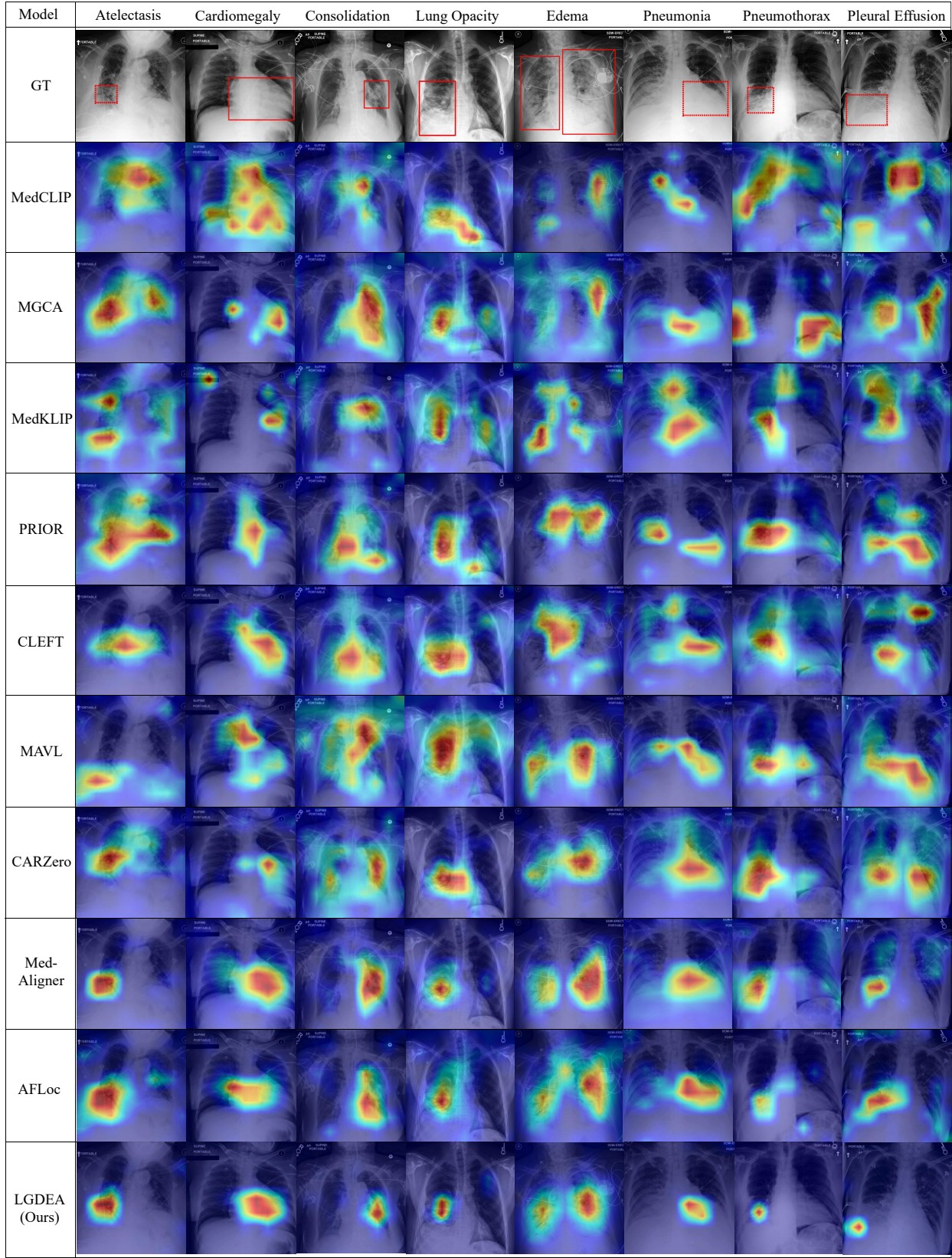

*Figure 7.* Attention maps for eight disease categories are visualized on the MS-CXR dataset, comparing LGDEA with nine baseline methods. Red bounding boxes indicate the ground truth regions relevant to phrase grounding. Highlighted pixels correspond to higher activation weights, reflecting stronger associations between specific diagnostic terms and image regions.

*Table 11.* The Image–Text Retrieval task is compared with the all ablated variants on the MIMIC-5 × 200 dataset, and Precision (%) scores are reported.

| Model | Prec@1 | Prec@2 | Prec@5 | Prec@10 |
|---|---|---|---|---|
| Full LGDEA | 56.31 | 53.85 | 53.71 | 50.65 |
| w/o $\mathcal{L}_{rec}$ | 51.56 | 50.42 | 48.12 | 45.29 |
| w/o $\mathcal{L}_{p}^{evid}$ | 54.06 | 52.12 | 49.89 | 46.37 |
| w/o $\mathcal{L}_{u}^{evid}$ | 53.12 | 51.34 | 50.24 | 47.68 |
| w/o $\mathcal{L}_{evi\text{-}align}$ | 53.25 | 51.68 | 50.47 | 48.61 |

choice of evidence extractor, we replace Spark-Desk with two representative open-source LLMs, Qwen-7B and LLaMA-8B, while keeping all other components and training settings unchanged. As shown in Tables 6 and 12, LGDEA achieves stable performance on both phrase grounding and image classification across different LLMs. Although different LLMs may produce diagnostic evidence with slightly different linguistic expressions and levels of granularity, the overall performance remains consistent. This indicates that LGDEA does not rely on a specific evidence extractor, but mainly benefits from the semantic relevance and clinical validity of the extracted diagnostic evidence. These results further demonstrate the robustness and practical applicability of LGDEA when different LLM-based evidence extraction tools are used.

*Table 12.* The performance of the phrase grounding task is compared with LLaMA and Qwen on the MS-CXR dataset, using Contrast-to-Noise Ratio (CNR) scores across eight disease categories.

| Model | Atelectasis | Cardiomegaly | Consolidation | Lung Opacity | Edema | Pneumonia | Pneumothorax | Pleural Effusion |
|---|---|---|---|---|---|---|---|---|
| LLaMA | 0.8000 | 0.9443 | 0.8343 | 0.9235 | 0.8115 | 0.9271 | 0.6894 | 0.9328 |
| Qwen | 0.8182 | 0.9469 | 0.8314 | 0.9160 | 0.8020 | 0.8805 | 0.8521 | 0.8003 |
| Ours | **0.8449** | **0.9525** | **0.8357** | **0.9564** | **0.8172** | **0.9353** | **0.8526** | **1.0928** |

