# OpenReview forum: "LLM-Guided Diagnostic Evidence Alignment for Medical Vision–Language Pretraining under Limited Pairing"
_ICML.cc/2026/Conference — ICML 2026 regular_

### Official Review · Reviewer_sW3P · 2026-03-02

**Soundness:** 2
**Presentation:** 2
**Significance:** 3
**Originality:** 3
**Overall Recommendation:** 4
**Confidence:** 4

**Summary:**

This paper proposes a new evidence-based paradigm for medical vision-language pretraining to address the issue that current models tend to over-focus on either global or local context. The authors carefully design a pipeline that uses limited paired data to learn cross-modal representations, while leveraging unpaired data to learn unimodal evidence embeddings. Without fine-grained supervision, the proposed method still achieves effective evidence grounding. Experiments under limited-pairing pretraining settings, including both in-domain and cross-domain evaluations, show consistent gains on different tasks. Ablations and comparisons across different LLMs further support the contribution of the key components.

**Compliance With Llm Reviewing Policy:**

Affirmed.

**Final Justification:**

The rebuttal has addressed my concern in the initial review, and I am glad to keep my score as weak accept.

**Key Questions For Authors:**

As discussed in the weaknesses, my concerns mainly focus on two points:
1. The availability of paired data is not a major practical challenge, so it is crucial to evaluate how the method performs when the paired-data ratio is higher.
2. A major contribution of the paper is that using only global image-text pairs can still train the model for lesion grounding. To support this claim, the paper needs broader comparisons with more methods and a stronger evaluation using additional metrics.

If these questions are not addressed effectively during the rebuttal, the overall assessment of the paper may decrease.

**Limitations:**

The proposed method is impressive, but it has two main limitations:
1. The practical effectiveness of the proposed "limited paired data" setting in real clinical workflows remains unclear.
2. Since the method does not rely on fine-grained supervision, its grounding accuracy is likely to be minor than specialized model

**Strengths And Weaknesses:**

Strength:
1. This paper introduces diagnostic evidence as a clinically meaningful alignment, which addresses a key issue in prior medical vision-language pretraining where models tend to over-focus on either global or local context.
2.The paper proposes a unique strategy that uses limited paired data to learn cross-modal representations, while using unpaired data to model unimodal evidence distributions, which makes effective use of limited supervision.
3.The proposed method shows strong performance across multiple downstream tasks in both single-domain and cross-domain settings, and the ablation studies further support the effectiveness of the approach.

Major Weakness:
1. The paper motivates the setup by arguing that paired medical images and radiology reports are limited. However, as also mentioned in Section 4.1 Training Data, MIMIC-CXR provides substantial paired data, so this does not seem to be the main practical challenge.
2. Since the comparisons are conducted on chest X-ray, the evaluation should include specialized chest X-ray methods, such as MAIRA-2, which supports both classification and lesion grounding.
3. The claim that lesion-level visual evidence can be effectively grounded without fine-grained annotations is interesting, but CNR is not a standard metric for grounding. Figure 4 is also limited as qualitative evidence. It is helpful to report additional commonly used grounding metrics, such as IoU, for a more convincing evaluation.
4. As noted in Weakness 1, if paired data is not a major bottleneck, it is important to analyze how performance changes when increasing the paired-data ratio in LGDEA, for example to 20% or 50%. This can clarify whether the method continues to scale with more paired supervision.

Minor Issues
1. Figure 2 can be moved to Page 3 to make the method easier to understand earlier in the paper

---

> ### Author Rebuttal · Authors · 2026-03-30
>
> We sincerely thank the reviewer for the thoughtful and constructive comments. We address each point below.
>
> **Major Weakness 1 / Key Question 1 / Limitation 1：**
>
> - **Clinical relevance.** In real clinical workflows, especially during the cold-start stage of a new hospital or department, high-quality image–report pairs directly usable for training are often limited. Although large amounts of historical imaging and text data may exist, cross-system integration issues and inconsistent labeling standards often prevent them from being readily used as paired data [1].
> - **Experimental design.** We use MIMIC-CXR, a large-scale chest X-ray image–report dataset, to validate our idea. To mimic the above scenario, we retain only 5% or 10% of the paired samples in MIMIC-CXR and treat the remaining images and reports as unpaired unimodal data. In the cross-domain setting, we further introduce CheXpert, a public image-only chest X-ray dataset, as additional unpaired visual data.
>
> **Major Weakness 2 / Key Question 2：**
>
> We further include MAIRA-2 for comparison on lesion localization and disease classification. Notably, MAIRA-2 is trained with about 2× more samples than LGDEA, uses box supervision, and adopts an image encoder pretrained on 1.4M chest X-ray images. As shown in Tables 1 and 2, while MAIRA-2 achieves stronger classification performance, LGDEA performs better on lesion localization, highlighting the advantage of evidence-level alignment for lesion localization.
>
> Table 1. RSNA lesion localization: LGDEA vs. MAIRA-2.
> | Model | IoU | Dice |
> |--------|-----:|-----:|
> | MAIRA-2 | 28.23 | 40.84 |
> | LGDEA | 39.65 | 53.20 |
>
> Table 2. Disease classification (ACC) on COVID, RSNA, and NIH14: LGDEA vs. MAIRA-2.
> | Model | COVID | RSNA | NIH14 |
> |--------|------:|-----:|------:|
> | MAIRA-2 | 96.93 | 81.07 | 88.88 |
> | LGDEA | 90.47 | 79.26 | 87.06 |
>
> **Major Weakness 3 / Key Question 2 / Limitation 2：**
>
> - **Stronger grounding evaluation.** We complement CNR and qualitative results with IoU and Dice to provide a more standard localization evaluation (Table 3).
>
> - **Fine-grained grounding ability of LGDEA.** By modeling textual evidence as soft combinations of prototypes and learning lesion-aware local visual evidence, LGDEA better aligns subtle clinical semantics with localized pathology under limited pairing.
>
> Table 3. Lesion localization on the RSNA dataset: comparison with baselines in terms of IoU and Dice.
> | Model | IoU | Dice |
> |------------|-----:|-----:|
> | MedCLIP | 24.96 | 34.67 |
> | GLoRIA | 33.30 | 46.17 |
> | BioViL | 22.16 | 35.81 |
> | MGCA | 35.52 | 48.65 |
> | MedKLIP | 31.39 | 46.05 |
> | PRIOR | 33.06 | 45.86 |
> | CLEFT | 34.90 | 47.87 |
> | MAVL | 35.44 | 51.02 |
> | CARZero | 36.12 | 49.48 |
> | AFLoc | 35.79 | 48.13 |
> | MedAligner | 37.92 | 51.28 |
> | MAIRA2 | 28.23 | 40.84 |
> | **LGDEA** | **39.65** | **53.20** |
>
> **Major Weakness 4 / Key Question 1：**
>
> **Performance under different pairing ratios.** Our work studies medical vision-language pretraining with limited image-report pairs. Results at 5% and 10% already show that LGDEA matches or even surpasses the fully paired setting on downstream tasks with only 10% paired data. To further assess scalability, we also include 20% and 50% pairing ratios in Tables 4–6, and the results show that performance improves as more paired data become available.
>
> Table 4. Phrase grounding under different pairing ratios.
> | Pairing Ratio | Atelectasis | Cardiomegaly | Consolidation | Lung Opacity | Edema | Pneumonia | Pneumothorax | Pleural Effusion |
> |----------|------------:|-------------:|--------------:|-------------:|-------:|----------:|-------------:|-----------------:|
> | 5% | 0.8000 | 0.9627 | 0.8071 | 0.9032 | 0.8081 | 0.7849 | 0.7902 | 0.9742 |
> | 10% | 0.8449 | 0.9525 | 0.8357 | 0.9564 | 0.8172 | 0.9353 | 0.8526 | 1.0928 |
> | 20% | 0.8502 | 0.9683 | 0.8833 | 0.9654 | 0.8257 | 0.9762 | 0.8609 | 1.0983 |
> | 50% | 0.8978 | 0.9893 | 0.9180 | 1.0286 | 0.8571 | 1.0045 | 0.9024 | 1.1349 |
>
> Table 5. Image–text retrieval under different pairing ratios.
> | Pairing Ratio | Prec@1 | Prec@2 | Prec@5 | Prec@10 |
> |---------------|-------:|-------:|-------:|--------:|
> | 5% | 53.25 | 51.03 | 50.66 | 49.98 |
> | 10% | 56.31 | 53.85 | 53.71 | 50.65 |
> | 20% | 57.43 | 55.14 | 54.63 | 52.19 |
> | 50% | 60.82 | 57.80 | 55.91 | 55.87 |
>
> Table 6. Image classification under different pairing ratios.
> | Pairing Ratio | MIMIC-5×200 | COVID | RSNA | NIH14 |
> |---------------|------------:|------:|-----:|------:|
> | 5% | 79.44 | 87.53 | 78.48 | 85.88 |
> | 10% | 80.00 | 90.47 | 79.26 | 87.06 |
> | 20% | 80.16 | 92.73 | 80.20 | 88.65 |
> | 50% | 82.68 | 94.43 | 81.35 | 90.04 |
>
> **Minor Issue：**
>
> We agree that moving Figure 2 earlier will improve readability, and we will revise the paper accordingly.
>
> [1] Ye, et al. Continual self-supervised learning: Towards universal multi-modal medical data representation learning. CVPR. 2024.

---

> > ### Author Rebuttal · Reviewer_sW3P · 2026-04-02
> >
> > The rebuttal has addressed my concern, and I am glad to keep my score.

---

> > > ### Author Response · Authors · 2026-04-02
> > >
> > > Dear Reviewer sW3P,
> > >
> > > Thank you for your positive feedback and for recognizing that we have addressed all of your concerns. We greatly appreciate your time and thoughtful review, which helped strengthen our paper.
> > >
> > > Sincerely,
> > >
> > > The Authors of Paper 20454

---

### Official Review · Reviewer_jofQ · 2026-03-05

**Soundness:** 3
**Presentation:** 4
**Significance:** 3
**Originality:** 3
**Overall Recommendation:** 5
**Confidence:** 5

**Summary:**

This paper tackles the challenge of cross-modal alignment under scarce paired data in medical vision–language pretraining and proposes the LGDEA framework. By extracting diagnostic evidence with LLMs and constructing a shared prototype space, the method aligns images and reports at the evidence level and extends supervision to unpaired data through relation propagation. Experiments show that LGDEA significantly outperforms existing methods under low pairing ratios and can rival models trained with fully paired data, demonstrating strong effectiveness and practical value.

**Compliance With Llm Reviewing Policy:**

Affirmed.

**Final Justification:**

The author's reply has resolved my doubts, and I recommend accepting this paper.

**Key Questions For Authors:**

1. Compared with standard CLIP-style medical vision–language pretraining methods, does the LLM-based diagnostic evidence extraction module introduce additional computational overhead?

2. Since the paper models cross-modal alignment from the perspective of diagnostic evidence, could the authors further discuss how this design corresponds to real clinical diagnostic workflows?

3. Will the pretrained model weights be released in the future? This could positively contribute to the advancement of research in medical multimodal pretraining.

**Limitations:**

yes

**Strengths And Weaknesses:**

Strengths:
1. The scarcity of high-quality paired data is a long-standing bottleneck in medical VLP. The motivation of this work aligns well with real clinical data scenarios and has high practical relevance.

2. The method elevates cross-modal alignment from the feature level to the diagnostic evidence level, which better reflects the real clinical diagnostic process.

3. Instead of simple text augmentation, the method leverages LLMs to extract structured diagnostic evidence and constructs a shared prototype space for cross-modal alignment. This design effectively enables large-scale unpaired data to participate in training and allows for a natural and effective integration of LLMs with pretraining.

4. The experimental setup includes multiple pairing ratios, single-domain and cross-domain settings, and evaluations on diverse downstream tasks. The experiments are comprehensive and provide convincing evidence of the method’s effectiveness.

Weaknesses:
1. It would be beneficial if the authors could further analyze the impact of diagnostic evidence extraction quality on overall performance, such as whether different LLMs or prompt strategies significantly affect the results.

2. The authors may consider including more recent works on leveraging unpaired multimodal data in the introduction or related work section, such as UML [1].

[1] Gupta, S., Sundaram, S., Wang, C., Jegelka, S., and Isola, P. Better together: Leveraging unpaired multimodal data for stronger unimodal models. In The Fourteenth International Conference on Learning Representations, 2026.

---

> ### Author Rebuttal · Authors · 2026-03-30
>
> We sincerely thank the reviewer for the constructive and helpful comments. We address each point below.
>
> **Weakness 1：**
>
> Thank you for this valuable suggestion. We have already examined the effect of different LLM-based evidence extractors in Section 4.3. The existing results show that replacing Spark-Desk with Qwen-7B or LLaMA-8B leads to largely consistent performance on both phrase grounding and classification, suggesting that LGDEA does not rely on a specific LLM, but rather on the quality of the learned diagnostic evidence space and evidence-level alignment.  To further evaluate sensitivity to evidence extraction quality, we additionally conduct prompt-sensitivity experiments using different prompt styles. As shown in Tables 1 and 2, LGDEA remains relatively stable under prompt variation. For completeness, we will include both the original prompt template and the additional prompt templates in the revised manuscript.
>
> Table 1 Phrase grounding using original prompt template and the additional prompt templates.
> | Variation | Atelectasis | Cardiomegaly | Consolidation | Lung Opacity | Edema | Pneumonia | Pneumothorax | Pleural Effusion |
> |-----------|------------:|-------------:|--------------:|-------------:|------:|----------:|-------------:|-----------------:|
> | Qwen_ori | 0.8182 | 0.9469 | 0.8314 | 0.9160 | 0.8020 | 0.8805 | 0.8521 | 0.8003 |
> | Qwen_add | 0.8276 | 0.9487 | 0.8465 | 0.9231 | 0.7573 | 0.8745 | 0.8624 | 0.8100 |
> | LGDEA_ori | 0.8449 | 0.9525 | 0.8357 | 0.9564 | 0.8172 | 0.9353 | 0.8526 | 1.0928 |
> | LGDEA_add | 0.8472 | 0.9746 | 0.8309 | 0.9343 | 0.8073 | 0.9412 | 0.8494 | 1.0324 |
>
> Table 2 Zero-shot classification using original prompt template and the additional prompt templates.
> | Variation | MIMIC-5×200 | COVID | RSNA | NIH14 |
> |-----------|------------:|------:|-----:|------:|
> | Qwen_ori | 79.92 | 90.39 | 79.15 | 86.14 |
> | Qwen_add | 79.88 | 90.25 | 78.97 | 86.35 |
> | LGDEA_ori | 80.00 | 90.47 | 79.26 | 87.06 |
> | LGDEA_add | 79.96 | 90.32 | 79.43 | 87.20 |
>
> **Weakness 2：**
>
> We agree that the manuscript should better position LGDEA within the broader literature on learning from unpaired multimodal data. More specifically, UML leverages unpaired multimodal data to improve unimodal representation learning through cross-modal parameter sharing, without relying on explicit correspondence or alignment inference. In contrast, LGDEA addresses medical vision-language pretraining under limited pairing, and aims to learn evidence-level cross-modal alignment through an LLM-guided diagnostic evidence space, lesion-level visual evidence learning, and relation propagation.
>
> **Question 1：**
>
> - **Offline extraction.** LGDEA does introduce an additional LLM-based diagnostic evidence extraction stage compared with standard CLIP-style medical vision-language pretraining methods. However, this step is performed offline and therefore does not incur repeated backpropagation overhead during training. The main extra training cost comes from prototype assignment, lesion-query attention, and in-batch relation propagation, while the image and text encoders themselves are not replaced by heavier backbones.
>
> - **Efficiency comparison.** To make this overhead more explicit, we further add a comparison with representative baselines under the same hardware setting, including training time, GPU usage, and inference time, as shown in Table 3. These results will be incorporated into the revised manuscript to provide a more comprehensive evaluation of the trade-off between computational cost and performance improvement.
>
> Table 3 Comparison with baselines in training time, GPU usage, and inference time on MIMIC-5×200, COVID, and RSNA.
> | Model | Training Time | GPU Usage | MIMIC-5×200 (s/image) | COVID (s/image) | RSNA (s/image) |
> |-------|--------------:|----------:|----------------------:|----------------:|---------------:|
> | GLoRIA | 15.3 | 31 | 0.104851 | 0.000106 | 0.000017 |
> | MedAligner | 16 | 33 | 0.083081 | 0.000089 | 0.000013 |
> | LGDEA | 13.4 | 31 | 0.004043 | 0.002749 | 0.002416 |
>
> **Question 2：**
>
> LGDEA is motivated by clinical reasoning: clinicians rely on key evidence, local lesion regions, and their integration. Accordingly, LGDEA aligns report-derived evidence with lesion-level visual evidence in a shared evidence space, and further strengthens this alignment through relation propagation under limited pairing.
>
> **Question 3：**
>
> We agree that releasing model weights is important. Due to anonymous review, we cannot release them now, but we plan to release the pretrained weights and key code if the paper is accepted.

---

> > ### Author Rebuttal · Reviewer_jofQ · 2026-04-02
> >
> > The author's reply has resolved my doubts, and I recommend accepting this paper.

---

> > > ### Author Response · Authors · 2026-04-02
> > >
> > > Dear Reviewer jofQ,
> > >
> > > Thank you for your positive evaluation. We also sincerely appreciate your recognition that we have clarified and addressed the concern. Your valuable feedback has helped improve the quality of this manuscript.
> > >
> > > Sincerely,
> > >
> > > The Authors of Paper 20454

---

### Official Review · Reviewer_RThY · 2026-03-10

**Soundness:** 4
**Presentation:** 4
**Significance:** 3
**Originality:** 3
**Overall Recommendation:** 5
**Confidence:** 5

**Summary:**

This paper proposes the LGDEA framework, which introduces diagnostic evidence as an intermediate semantic layer for cross-modal alignment in medical vision–language pretraining. By leveraging LLMs to extract key diagnostic evidence and constructing a shared prototype space, the method aligns image lesions with textual evidence and extends supervision from limited paired data to abundant unpaired data through relation propagation. Experiments show that LGDEA achieves strong performance even with very low pairing ratios, demonstrating the effectiveness of evidence-level alignment.

**Compliance With Llm Reviewing Policy:**

Affirmed.

**Final Justification:**

My concerns have now been resolved, and I maintain my positive rating.

**Key Questions For Authors:**

1. Could the authors further discuss the adaptability of LGDEA to diseases or clinical scenarios that are not explicitly covered in the training data?

2. From a representation learning perspective, could the authors elaborate on the fundamental differences between evidence-level alignment and traditional CLIP-style alignment in terms of optimization objectives and representation structures?

**Limitations:**

Yes.

**Strengths And Weaknesses:**

Strengths:

1. The paper introduces a novel evidence-level alignment perspective, shifting from feature-level alignment to diagnostic evidence alignment, which has profound implications for medical multimodal pretraining.

2. The proposed method is well-designed and logically coherent. It constructs a diagnostic prototype space via LLM-based evidence extraction, learns lesion-level visual evidence, and establishes weakly supervised relation propagation to effectively utilize large-scale unpaired data for cross-modal alignment.

3. The experiments cover multiple pairing ratios as well as both single-domain and cross-domain settings. Comprehensive ablation and hyperparameter studies are conducted, and the experimental results thoroughly validate the effectiveness of the proposed method.

4. The method surpasses other approaches trained with full paired data using only 10% paired samples, demonstrating significant practical value in real-world medical scenarios.

Weaknesses:

1. Although the paper evaluates the impact of different LLMs on performance, it would be beneficial to further analyze the sensitivity of the model to variations in prompt quality or evidence extraction quality.

2. It would be helpful if the authors could further discuss whether the proposed framework can be extended to more general domains or other multimodal tasks.

3. The introduction of diagnostic evidence space modeling and cross-modal relation propagation may incur additional computational overhead. The authors could consider providing a brief comparison with baseline models in terms of training time and memory consumption to facilitate a more comprehensive evaluation of the method.

---

> ### Author Rebuttal · Authors · 2026-03-30
>
> We sincerely thank the reviewer for the thoughtful and constructive comments. We address each point below.
>
> **Weakness 1：**
>
> - **Different LLM Extractors.** As shown in Section 4.3, LGDEA remains stable across different LLM-based evidence extractors (e.g., Spark and Qwen), indicating that it does not rely on any specific LLM but mainly benefits from the representation of extracted diagnostic evidence.
>
> - **Prompt Sensitivity.** To further assess sensitivity to prompt quality, we add prompt-sensitivity experiments with different prompt styles. The results in Tables 1,2 show that LGDEA is robust to prompt variation. We will include both the original and additional prompt templates in the revised manuscript.
>
> Table 1 Phrase grounding using original prompt template and the additional prompt templates.
> | Variation | Atelectasis | Cardiomegaly | Consolidation | Lung Opacity | Edema | Pneumonia | Pneumothorax | Pleural Effusion |
> |-----------|------------:|-------------:|--------------:|-------------:|------:|----------:|-------------:|-----------------:|
> | Qwen_ori | 0.8182 | 0.9469 | 0.8314 | 0.9160 | 0.8020 | 0.8805 | 0.8521 | 0.8003 |
> | Qwen_add | 0.8276 | 0.9487 | 0.8465 | 0.9231 | 0.7573 | 0.8745 | 0.8624 | 0.8100 |
> | LGDEA_ori | 0.8449 | 0.9525 | 0.8357 | 0.9564 | 0.8172 | 0.9353 | 0.8526 | 1.0928 |
> | LGDEA_add | 0.8472 | 0.9746 | 0.8309 | 0.9343 | 0.8073 | 0.9412 | 0.8494 | 1.0324 |
>
> Table 2 Zero-shot classification using original prompt template and the additional prompt templates.
> | Variation | MIMIC-5×200 | COVID | RSNA | NIH14 |
> |-----------|------------:|------:|-----:|------:|
> | Qwen_ori | 79.92 | 90.39 | 79.15 | 86.14 |
> | Qwen_add | 79.88 | 90.25 | 78.97 | 86.35 |
> | LGDEA_ori | 80.00 | 90.47 | 79.26 | 87.06 |
> | LGDEA_add | 79.96 | 90.32 | 79.43 | 87.20 |
>
> **Weakness 2：**
>
> LGDEA is not specific to chest X-rays and reports, but is based on a more general idea: one modality provides semantic evidence, the other provides local perceptual evidence, and the two can be softly aligned in a shared evidence space. With textual evidence extraction, shared evidence learning, and intra-modal graph propagation, LGDEA may extend to other medical image-text tasks and more general multimodal settings with sparse cross-modal pairing.
>
> **Weakness 3：**
>
> Although LGDEA introduces an LLM-based evidence extraction stage, this step is performed offline and does not incur repeated training-time backpropagation cost. The main overhead comes from prototype assignment, lesion-query attention, and relation propagation. We therefore report training time, GPU usage, and inference time in Table 3 for a fair efficiency comparison.
>
> Table 3 Comparison with baselines in training time, GPU usage, and inference time on MIMIC-5×200, COVID, and RSNA.
> | Model | Training Time | GPU Usage | MIMIC-5×200 (s/image) | COVID (s/image) | RSNA (s/image) |
> |-------|--------------:|----------:|----------------------:|----------------:|---------------:|
> | GLoRIA | 15.3 | 31 | 0.104851 | 0.000106 | 0.000017 |
> | MedAligner | 16 | 33 | 0.083081 | 0.000089 | 0.000013 |
> | LGDEA | 13.4 | 31 | 0.004043 | 0.002749 | 0.002416 |
>
> **Question 1：**
>
> We would like to clarify that the primary focus of LGDEA is medical vision-language pretraining under limited correspondence, rather than generalization to disease categories or clinical scenarios that are entirely outside the training set. In our setting, the main challenge is how to better exploit limited paired image–report samples together with abundant unpaired images and reports. That said, we agree that the adaptability of LGDEA to diseases or clinical scenarios not explicitly covered in training is an interesting and important question. We agree this is an important direction and will discuss it in the revised manuscript.
>
> **Question 2：**
>
> - **Objective.**  Traditional CLIP-style methods optimize feature-level alignment by directly maximizing the similarity between global image and report embeddings. In contrast, LGDEA optimizes evidence-level alignment: it first constructs a shared diagnostic evidence space from report evidence, then aligns lesion-level visual evidence with report-derived evidence, and further leverages propagated cross-modal relations as weak supervision under limited pairing.
>
> - **Representation.**  Accordingly, the learned representations are also different. CLIP-style alignment mainly relies on a shared global embedding space. LGDEA instead introduces diagnostic prototypes, lesion queries, and evidence-centric representations, so that images and reports are organized around latent diagnostic evidence rather than only global feature similarity.

---

> > ### Author Rebuttal · Reviewer_RThY · 2026-04-02
> >
> > Thank you for the authors‘ response. My concerns have now been resolved.

---

> > > ### Author Response · Authors · 2026-04-02
> > >
> > > Dear Reviewer RThY,
> > >
> > > Thank you very much for your positive evaluation and for recognizing that we have addressed your concerns. Your feedback has helped improve the quality and clarity of our manuscript.
> > >
> > > Sincerely,
> > >
> > > The Authors of Paper 20454

---

### Official Review · Reviewer_m596 · 2026-03-12

**Soundness:** 2
**Presentation:** 3
**Significance:** 3
**Originality:** 2
**Overall Recommendation:** 3
**Confidence:** 4

**Summary:**

The manuscript proposes LGDEA, a medical VLM pretraining method for settings where only a small fraction of chest X-ray images and radiology reports are paired, while many additional images and reports are unpaired. The main claim is that this evidence-level alignment better reflects clinical representation and reduces dependence on paired data, and the experiments provided show improved phrase grounding, image-text retrieval, and zero-shot classification compared with prior baselines. Overall, the idea is novel and interesting with some doubts around the experimental design and evaluation scheme.

**Compliance With Llm Reviewing Policy:**

Affirmed.

**Final Justification:**

With the updated baseline, some of my concerns have been resolved. However, the concern around the inability to represent textual signals, such as negation, remains, as the authors empirically showed it during the discussion phase that the performance drops for more complex text.

**Key Questions For Authors:**

1. In the proposed method, the author practivcally limit the evidence set to a finite space of $k$ evidence. I wonder how the author think about the outlier or rare sample in the training set. This practically eliminates them from being present in the dictionary of evidence and biases the model to the majority evidence.

**Limitations:**

The work needs a more broad limitaion discussion around the model failure points.

**Strengths And Weaknesses:**

I have read the paper, and the work is presented clearly and tries to answer a clinically important question. Even though the work is somewhat original in my opinion, I have three main concerns regarding the soundness of the claims and the significance of the work as listed below:

- In Lines 151–152, the authors claim that textual prototype learning helps identify “a concise and clinically meaningful abnormal finding.” I am not fully convinced this claim is sufficiently supported. Learning prototypes in textual latent space may struggle to distinguish important semantic factors such as negation, uncertainty, chronicity, comparison statements, and anatomical modifiers. For example, expressions such as “X specified as Y” and “X is not specified as Y” can remain geometrically very close in representation space despite having very different clinical meanings. This limitation does not appear to be explicitly addressed in the method design or analysis.

- The experimental setting is effectively one of learning from paired and unpaired multimodal data, where cross-modal correspondence is missing for a large portion of samples. This is closely related to prior work on missing-modality learning, incomplete multimodal learning, missing-view learning, and semi-paired multimodal representation learning. However, the related work is framed primarily through the lens of medical global/local alignment. I encourage the authors to position the paper more directly with respect to these related areas and to include a sufficient set of relevant baselines from that literature. Such example works can be (not limited to) found in [1].

- The training experiments rely on a random selection of the paired subset, which raises concerns about the reliability and stability of the reported results. Under this setup, performance may depend materially on the specific sampled split. At minimum, the conclusions would be more convincing if supported by repeated runs or a 5-fold cross-validation style protocol to demonstrate that the reported gains reflect a stable and meaningful signal rather than random sampling/chance.

[1] Wu, R., Wang, H., Chen, H. T., & Carneiro, G. Deep Multimodal Learning with Missing Modality: A Survey. Transactions on Machine Learning Research.

---

> ### Author Rebuttal · Authors · 2026-03-30
>
> We sincerely thank the reviewer for the thoughtful and constructive comments. We address each point below.
>
> **Weakness 1:**
>
> - **Scope and nuanced semantics.** We agree that subtle clinical semantics, such as negation, uncertainty, comparison, and anatomical modifiers, are challenging and not the primary focus of LGDEA, which targets medical vision-language pretraining under limited correspondence. Nevertheless, LGDEA uses soft prototype assignment and prototype-based reconstruction, allowing each evidence phrase to be represented by multiple prototypes and thus helping capture subtle semantic differences. We verify this with a semantic perturbation retrieval experiment on MIMIC-5x200, where hard variants are constructed by altering negation, uncertainty, anatomy, and comparison. Table 1 shows that LGDEA remains robust under these perturbations, indicating its ability to distinguish subtle clinical semantics.
>
> Table 1 Hard-negative image-to-text retrieval on MIMIC-5×200.
> | Disease | Top-1 ACC | MRR | Positive | Negation | Uncertainty | Anatomy | Comparison |
> |------------------|----------:|-------:|---------:|-------:|----------:|---------:|--------:|
> | Atelectasis | 0.5650 | 0.6947 | 0.5650 | 0.2600 | 0.0000 | 0.1700 | 0.0050 |
> | Cardiomegaly | 0.6142 | 0.7271 | 0.6142 | 0.2995 | 0.0051 | 0.0761 | 0.0051 |
> | Consolidation | 0.6935 | 0.7953 | 0.6935 | 0.2111 | 0.0000 | 0.0905 | 0.0050 |
> | Edema | 0.6800 | 0.7695 | 0.6800 | 0.2650 | 0.0000 | 0.0550 | 0.0000 |
> | Pleural Effusion | 0.6350 | 0.7403 | 0.6350 | 0.2750 | 0.0000 | 0.0900 | 0.0000 |
> | Average | 0.6375 | 0.7454 | 0.6375 | 0.2621 | 0.0010 | 0.0963 | 0.0030 |
>
> **Weakness 2:**
>
> - **Problem relation and additional baseline.** We agree that LGDEA is better framed as incomplete multimodal learning rather than conventional missing-modality learning, since the main challenge is missing cross-modal correspondence across much of the dataset rather than missing modalities within paired samples. We will revise the related work accordingly and add DrFuse as a representative medical incomplete multimodal learning baseline. DrFuse is trained under the same MIMIC-CXR setting as LGDEA. As DrFuse does not support fine-grained grounding, we compare it only on retrieval and classification (Table 2).
>
> Table 2 LGDEA vs. DrFuse on MIMIC-5×200 retrieval and zero-shot classification (ACC) on MIMIC-5×200, COVID, RSNA, and NIH14.
> | Method | Prec@1 | Prec@2 | Prec@5 | Prec@10 |MIMIC-5×200 | COVID | RSNA | NIH14 |
> |--------|-------:|-------:|-------:|--------:|--------:|--------:|--------:|--------:|
> | Drfuse | 50.71 | 49.13 | 48.85 | 46.73 |73.79 | 82.81 | 73.49 | 74.82 |
> | LGDEA | 56.31 | 53.85 | 53.71 | 50.65 |80.00 | 90.47 | 79.26 | 87.06 |
>
> **Weakness 3:**
>
> We agree and therefore conduct repeated-resampling experiments at a fixed pairing ratio (e.g., 10%) over 10 independent runs and report mean±std on phrase grounding, image–text retrieval, and zero-shot classification (Tables 3-5). Results show LGDEA achieves stable gains that are not due to favorable sampling splits.
>
> Table 3 Phrase grounding stability results (CNR).
> | Method | Atelectasis | Cardiomegaly | Consolidation | Lung Opacity | Edema | Pneumonia | Pneumothorax | Pleural Effusion |
> |--------|------------:|-------------:|--------------:|-------------:|------:|----------:|-------------:|-----------------:|
> | LGDEA | 0.8353±0.0680 | 0.9647±0.0135 | 0.8407±0.0526 | 0.9459±0.0447 | 0.8211±0.0353 | 0.9244±0.0338 | 0.8548±0.0495 | 1.0393±0.0813 |
>
> Table 4 Image-text retrieval stability results (Precision).
> | Method | Prec@1 | Prec@2 | Prec@5 | Prec@10 |
> |--------|-------:|-------:|-------:|--------:|
> | LGDEA | 56.43±0.0735 | 53.98±0.0068 | 53.49±0.0127 | 50.74±0.1038 |
>
> Table 5 Zero-shot classification stability results (ACC).
> | Method | MIMIC-5×200 | COVID | RSNA | NIH14 |
> |--------|------------:|------:|-----:|------:|
> | LGDEA | 80.16±0.0314 | 90.80±0.0164 | 79.04±0.0188 | 87.12±0.0126 |
>
> **Key Question:**
>
> - **Soft prototype assignment.** We clarify that the prototype space in LGDEA is not a hard evidence dictionary that removes rare samples. Instead, each evidence phrase is represented by soft assignment over multiple prototypes, while evidence-guided relation propagation further provide supervision through semantically related neighbors.
>
> - **Scope and outlook.** Our focus is medical vision-language pretraining under limited correspondence rather than long-tail optimization. The prototype space is learned in a data-driven manner as a compact semantic basis for evidence-level alignment; incorporating prior knowledge of rare diseases or rare evidence may further improve this component, which we will briefly discuss in the revision.
>
> **Limitation:**
>
> We will clarify that LGDEA is designed for medical vision-language pretraining under limited correspondence rather than fully unpaired multimodal learning, and will briefly discuss rare/long-tail evidence modeling and fully unpaired pretraining as future directions.

---

> > ### Author Rebuttal · Reviewer_m596 · 2026-04-02
> >
> > Thank you for providing the details and further clarification.
> >
> > 1. I have a hard time trying to compare Table 1 in the rebuttal with the results in the manuscript. Would you be able to point me to the correct table in the manuscript that is directly comparable to the new table? How should I compare the columns in the Table 1 in your response? And also, can you clarify what the metric is: Accuracy or Precision?
> >
> > 2. In case of revising the paper to match the missing or incomplete modality literature, I think the authors need to update their benchmark with enough representation from that space. I understand the author included one method in the rebuttal in a short period of time, but the scope of comparison needs to be fairly updated, expansive enough, and directly comparable with the correct literature.

---

> > > ### Author Response · Authors · 2026-04-04
> > >
> > > # Response to the Final Justification.
> > >
> > > We regret to see your Final Justification. Due to our reasons, you have been misled, and our efforts over the past year may have been in vain. We provide a revised interpretation of Table 1 in the rebuttal as follows:
> > >
> > > **(1) Clarification of Table 1.**
> > >
> > > - **Interpretation of Table 1.** Table 1 evaluates whether the model can distinguish the ground-truth report from four semantically perturbed alternatives. Top-1 ACC denotes the proportion of samples for which the ground-truth report is ranked first, and MRR reflects its average ranking quality. The five “Pred as …” columns indicate which candidate type is ranked first and therefore sum to 1. The first five rows report results for five diseases—Atelectasis, Cardiomegaly, Consolidation, Edema, and Pleural Effusion, and the last row gives the average across diseases. On average, LGDEA achieves 63.75% Top-1 ACC, well above the 20% accuracy expected under random guessing among five candidates. These results suggest that LGDEA is effective for fine-grained clinically relevant semantic discrimination.
> > > - **Protocol difference from manuscript Table 2.** Although both experiments are conducted on MIMIC-5×200, rebuttal Table 1 is not directly comparable to manuscript Table 2 because the evaluation protocols are different. Table 2 measures precision-based retrieval performance over 1,000 report candidates per image, whereas Table 1 evaluates semantic discrimination with a candidate set of five reports, including one ground-truth report and four Spark-LLM-generated semantic variants.
> > >
> > > Table 1. Hard-negative image-text retrieval on MIMIC-5×200.
> > > | Disease | Top-1 ACC | MRR | Pred as Positive | Pred as Negation | Pred as Uncertainty | Pred as Anatomy | Pred as Comparison |
> > > |------------------|----------:|-------:|---------:|-------:|----------:|---------:|--------:|
> > > | Atelectasis | 0.5650 | 0.6947 | 0.5650 | 0.2600 | 0.0000 | 0.1700 | 0.0050 |
> > > | Cardiomegaly | 0.6142 | 0.7271 | 0.6142 | 0.2995 | 0.0051 | 0.0761 | 0.0051 |
> > > | Consolidation | 0.6935 | 0.7953 | 0.6935 | 0.2111 | 0.0000 | 0.0905 | 0.0050 |
> > > | Edema | 0.6800 | 0.7695 | 0.6800 | 0.2650 | 0.0000 | 0.0550 | 0.0000 |
> > > | Pleural Effusion | 0.6350 | 0.7403 | 0.6350 | 0.2750 | 0.0000 | 0.0900 | 0.0000 |
> > > | Average | 0.6375 | 0.7454 | 0.6375 | 0.2621 | 0.0010 | 0.0963 | 0.0030 |
> > >
> > > **(2) Incomplete-modality baselines.**
> > >
> > > - **Expanded positioning in related work.** We agree that adding only one extra baseline in the rebuttal is insufficient. In the revision, we will add a related-work subsection on *Medical Multimodal Learning with Incomplete Data* and discuss representative methods, including MedCLIP [1], DrFuse [2], MedRAT [3], and CRL-MMNAR [4].
> > > - **Expanded benchmark comparison.** We further include these methods in the benchmark and evaluate them on phrase grounding, disease classification, and image-text retrieval (Tables 2–4). Across these results, LGDEA achieves competitive performance, which better demonstrates its advantage: modeling clinically meaningful diagnostic evidence under limited image-report correspondence, rather than relying only on missing-modality completion or feature fusion.
> > >
> > > Table 2. Phrase grounding comparison on MIMIC-5×200.
> > > | Method | Atelectasis | Cardiomegaly | Consolidation | Lung Opacity | Edema | Pneumonia | Pneumothorax | Pleural Effusion |
> > > |-----------|------------:|-------------:|--------------:|-------------:|------:|----------:|-------------:|-----------------:|
> > > | MedCLIP | 0.5554 | 0.5253 | 0.5767 | 0.5599 | 0.4433 | 0.6175 | 0.4686 | 0.5076 |
> > > | MedRAT | 0.5603 | 0.5648 | 0.6192 | 0.6385 | 0.5624 | 0.6408 | 0.5512 | 0.6842 |
> > > | DrFuse | 0.4318 | 0.4349 | 0.3972 | 0.3467 | 0.4174 | 0.4905 | 0.4298 | 0.3819 |
> > > | CRL-MMNAR | 0.5237 | 0.5750 | 0.4895 | 0.5834 | 0.5708 | 0.6517 | 0.5214 | 0.6269 |
> > > | LGDEA | 0.8449 | 0.9525 | 0.8357 | 0.9564 | 0.8172 | 0.9353 | 0.8526 | 1.0928 |
> > >
> > > Table 3. Disease classification comparison on COVID, RSNA, and NIH14.
> > > | Method | MIMIC-5×200 | COVID | RSNA | NIH14 |
> > > |-----------|------------:|------:|-----:|------:|
> > > | MedCLIP | 52.50 | 77.70 | 79.90 | 58.84 |
> > > | MedRAT | 74.35 | 78.42 | 72.87 | 74.69 |
> > > | DrFuse | 73.79 | 82.81 | 73.49 | 74.82 |
> > > | CRL-MMNAR | 74.27 | 84.63 | 74.58 | 78.31 |
> > > | LGDEA | 80.00 | 90.47 | 79.26 | 87.06 |
> > >
> > > Table 4. Image-text retrieval comparison on MIMIC-5×200.
> > > | Method | Prec@1 | Prec@2 | Prec@5 | Prec@10 |
> > > |-----------|-------:|-------:|-------:|--------:|
> > > | MedCLIP | 44.40 | 43.45 | 44.50 | 45.58 |
> > > | MedRAT | 50.28 | 50.61 | 48.70 | 47.16 |
> > > | DrFuse | 50.71 | 49.13 | 48.85 | 46.73 |
> > > | CRL-MMNAR | 50.96 | 50.17 | 48.92 | 47.39 |
> > > | LGDEA | 56.31 | 53.85 | 53.71 | 50.65 |
> > >
> > > [1] MedCLIP. EMNLP. 2022.
> > >
> > > [2] DrFuse. AAAI. 2024.
> > >
> > > [3] MedRAT. ECCV. 2024.
> > >
> > > [4] CRL-MMNAR. EMNLP. 2025.

---

### Decision · Program_Chairs · 2026-04-30

**Decision:**

Accept (regular)

**Comment:**

The manuscript proposes a medical vision–language pretraining framework designed for scenarios with limited paired multimodal data and abundant unpaired data, by introducing diagnostic evidence as a prototype for cross-modal alignment. Specifically, the method leverages LLMs to extract structured diagnostic evidence, constructs a shared prototype space, and propagates relations to extend supervision from paired to unpaired data, achieving good performance on tasks such as phrase grounding, retrieval, and zero-shot classification, even at low pairing ratios. The proposed idea is interesting, to shift from feature-level to clinically meaningful evidence-level alignment by integrating LLMs, and comprehensive experiments across diverse settings that validate effectiveness. However, reviewers raise the questions about insufficient positioning with respect to broader missing-modality learning literature, ability to distinguish important semantic factors in diagnostic evidence such as handling negation counterparts, as well as the sensitivity of the model to variations in prompt quality or evidence extraction quality.

The author provides an exhaustive response to address these questions with additional experiments including more baseline comparison and varying parameters. After the rebuttal and discussion, three reviewers have confirmed their concerns have been fully resolved. One reviewer has the remaining concern whether the prototype learning on textual features can be able to create meaningful evidence on negated or uncertain clinical text. The additional results (Table 1) in the rebuttal looks providing some promising signals where the majority of samples can predict the positive pair while the negative perturbation absolutely makes this task more challenging. It would be helpful if the author can give a few specific examples in the paper to further analyze such cases. Overall, the paper proposes an interesting and reasonable idea supported by promising empirical evidence, which can bring some values to the community. Thus, I recommend Accept.